# Respecting the limit:
# Bayesian optimization with a bound on the optimal value

**Hanyang Wang**                                                   *Hanyang.Wang@warwick.ac.uk*
*Warwick Mathematics Institute*
*University of Warwick*
*Coventry, CV4 7AL, United Kingdom*

**Juergen Branke**                                                 *Juergen.Branke@wbs.ac.uk*
*Warwick Business School*
*University of Warwick*
*Coventry, CV4 7AL, United Kingdom*

**Matthias Poloczek**                                              *matpol@amazon.com*
*Amazon*
*San Francisco, CA 94105, USA*

**Reviewed on OpenReview:** *https://openreview.net/forum?id=y5HfOotJLk*

## Abstract

In many real-world optimization problems, we have prior information about what objective function values are achievable. In this paper, we study the scenario that we have either exact knowledge of the minimum value or a, possibly inexact, lower bound on its value. We propose *bound-aware Bayesian optimization* (BABO), a Bayesian optimization method that uses a new surrogate model and acquisition function to utilize such prior information. We present SlogGP, a new surrogate model that incorporates bound information and adapts the Expected Improvement (EI) acquisition function accordingly. Empirical results on a variety of benchmarks demonstrate the benefit of taking prior information about the optimal value into account, and that the proposed approach significantly outperforms existing techniques. Furthermore, we notice that even in the absence of prior information on the bound, the proposed SlogGP surrogate model still performs better than the standard GP model in most cases, which we explain by its larger expressiveness.

## 1 Introduction

For many real-world black-box optimization problems, evaluating a solution can be computationally expensive, and optimization algorithms thus need to be sample efficient. Bayesian optimization (BO) is a global optimization method suitable for such problems (Brochu et al., 2010; Shahriari et al., 2015; Garnett, 2023). Due to its data efficiency, it is widely used in many areas, such as protein design (Stanton et al., 2022), chemistry (Folch et al., 2022), robotics design (Calandra et al., 2016) and hyperparameter tuning (Cho et al., 2020).

While BO typically assumes the objective function $f(\mathbf{x})$ to be a black-box, in some real-world applications, additional information about the achievable optimal value is available. For example, in hyperparameter tuning, the error rate of a model is always larger than or equal to 0%. In physical experiments, the temperature must be greater than or equal to $-273.16°C$, and the electric resistance must be greater than or equal to $0 \ \Omega$. Any heat engine, the efficiency cannot exceed the Carnot efficiency, which depends on the temperatures of the heat source and sink. The upper bound of data rate in a communication channel is given by the Shannon-Hartley theorem, which defines the maximum capacity based on bandwidth and signal-to-noise

ratio. Latency cannot be reduced below the physical propagation time of a signal. And in aerodynamics the drag cannot be reduced below zero because drag represents resistance.

Intuitively, exploiting such information should be helpful. The first BO paper that takes such information into account is Hutter et al. (2009), who point out that a log transformation of positive functions is usually beneficial. Other recent works that may take output bound information into consideration are Nguyen & Osborne (2020); Wang et al. (2018); Nguyen et al. (2021); Jeong & Kim (2021).

In this paper, we focus on BO for minimization with a known lower bound $f^b$ on the unknown optimal function value $f^*$, i.e., $f^* \geq f^b$. We propose a new algorithm, *bound-aware Bayesian optimization* (BABO), which makes use of this lower bound information to improve efficiency in BO. BABO is based on a novel surrogate model, *Shifted Logarithmic Gaussian Process*, or SlogGP, which can take into account prior information on a lower bound of the objective function. SlogGP is defined as $f(\mathbf{x}) = e^{g(\mathbf{x})} - \zeta$, where $g(\mathbf{x})$ is a GP and $\zeta$ is a learnable parameter. As the resulting predictive distribution is no longer normal, we adapt the well-known expected improvement (EI) acquisition function to SlogGP, resulting in SlogEI. We then show how SlogEI can be further modified to also take into account the information on the lower bound of the objective function. We call this new acquisition function *Shifted Logarithmic Truncated Expected Improvement* (SlogTEI). Combining SlogGP and SlogTEI, we obtain BABO.

We find that SlogGP with SlogEI can outperform a standard GP with EI in BO even if there is no information on the bounds of the objective function, primarily due to its larger expressiveness, where expressiveness refers to the ability of a model to represent a wide range of functions or patterns in the data. Incorporating lower bound knowledge into SlogGP and using SlogTEI (BABO) can further enhance its performance. We evaluate the proposed framework on several synthetic functions as well as three real-world applications. Empirical results demonstrate that our new method outperforms conventional BO and other algorithms designed for the case of a known lower bound.

The structure of this paper is as follows. Section 2 gives an introduction to Bayesian optimization and surveys the literature on BO with lower bound information. In Section 3, we introduce the SlogGP model and the corresponding acquisition function in the known-bound as well as the unknown-bound case. SlogGP and SlogTEI together form BABO, our bound aware Bayesian optimization. We also prove that SlogGP is more flexible than GP, which can explain the superior performance of SlogGP over GP even in the absence of lower bound information. Section 4 reports on the experimental results, demonstrating the empirical advantage of using BABO when a lower bound is known. The paper concludes with a summary and some avenues for future work.

## 2 Preliminaries

### 2.1 Bayesian Optimization

Bayesian optimization (BO) is a sequential strategy for global optimization of black-box functions. Given an objective function $f(\mathbf{x})$ and the feasible set $\mathcal{X}$, the goal of BO is to find an optimal solution $\mathbf{x}^* \in \text{argmin}_{\mathbf{x} \in \mathcal{X}} f(\mathbf{x})$.

BO consists of two main steps. The first step is to build a surrogate model based on historical observations $\{(\mathbf{x}_1, y_1), ..., (\mathbf{x}_N, y_N)\}$. A common choice for the surrogate model is a GP, though other models have been proposed, including random forest (Hutter et al., 2011), deep neural network (Snoek et al., 2015) and Mondrian trees (Wang et al., 2018). For more information on Gaussian processes, we refer to Rasmussen & Williams (2006) and Schulz et al. (2018).

The second step involves using the surrogate model for selecting the solution to be evaluated next, balancing the mean and variance predicted by the surrogate model, which is known as the exploration-exploitation trade-off. In BO, this balance is achieved by optimizing a so-called acquisition function $\alpha$. A common choice is Expected Improvement (EI) (Mockus, 1998). The information from the newly evaluated solution is then added to the set of observations and the surrogate model is updated before the next iteration.

## 2.2 Acquisition functions for $f^* \geq f^b$

If we have prior information that the global minimum $f^* := \min_x f(x)$ is at least $f^b$, then we can try to adjust the acquisition function to make use of the information. For Max-value Entropy Search (MES; Wang & Jegelka (2017)), this is straightforward and has already been used as baseline in Nguyen & Osborne (2020) and Wang et al. (2018). MES selects the next solution to evaluate where it expects the largest reduction in entropy of the predicted distribution of the optimal objective value. It does not assume prior knowledge about the optimal value, but uses Gumbel Sampling to sample a realization of $f^*$ from the posterior. However, according to Wang et al. (2018), given a lower bound $f^b$ on the minimum value $f^*$, we may calculate MES as $\alpha^{\mathrm{MES}^b}\left(\mathbf{x} \mid f^b\right) = \frac{\gamma\left(\mathbf{x}, f^b\right) \phi\left[\gamma\left(\mathbf{x}, f^b\right)\right]}{2\Phi\left(\gamma\left(\mathbf{x}, f^b\right)\right)} - \log \Phi\left(\gamma\left(\mathbf{x}, f^b\right)\right)$, where $\gamma\left(\mathbf{x}, f^b\right) = \frac{\mu(\mathbf{x}) - f^b}{\sigma(\mathbf{x})}$, $\phi(\cdot)$ and $\Phi(\cdot)$ are the PDF and CDF of the standard normal distribution. Note that this analytical expression for MES does not require Monte Carlo sampling. In addition, we show in Appendix A.1 that $\mathrm{MES}^b$ shares the same maximizer with the acquisition function $\mathbb{P}(f(\mathbf{x}) < f^b)$, i.e. $\operatorname{argmax}_{\mathbf{x} \in \mathcal{X}} \alpha^{\mathrm{MES}^b}\left(\mathbf{x} \mid f^b\right) = \operatorname{argmax}_{\mathbf{x} \in \mathcal{X}} \mathbb{P}(f(\mathbf{x}) < f^b)$.

## 2.3 Models for $f^* \geq f^b$

Besides in acquisition functions, we can attempt to incorporate the information into surrogate models.

Hutter et al. (2009; 2011) find that employing log-transformation as a preprocessing step for positive-valued functions, combined with a correspondingly adapted EI acquisition function, can enhance the performance of BO. When shifting the function to be positive given the information about the lower bound, this is similar to our SlogGP model with a fixed rather than learnable $\zeta$, and we will later show that it performs significantly worse than our model. Jeong & Kim (2021) propose the objective bound conditional Gaussian process (OBCGP). OBCGP introduces a parameter $\mathbf{x}_M$, and conducts Gaussian process regression conditioned on $(\mathbf{x}_M, f(\mathbf{x}_M))$ through variational inference. In cases where the lower bound $f^b$ is known, we can enforce the distribution of $f(\mathbf{x}_M)$ to adhere to this known bound. Nguyen & Osborne (2020) focus on the case when the exact value of the optimum $f^*$ is known, which can be viewed as a special case of known bound. A parabolic Gaussian process model $\frac{1}{2} g^2(\mathbf{x}) - f^*$ (Gunter et al., 2014) is used. To guarantee an analytical form of the posterior distribution, a linearization is applied, so the final model is $-\frac{1}{2}\mu_g^2(\mathbf{x}) + \mu_g(\mathbf{x}) g(\mathbf{x}) - f^*$. It should be noted that while the range of the parabolic Gaussian process is $[f^b, \infty)$, after linearization, the model becomes a GP and samples no longer adhere to the bound. Second, the transformation of the GP inflates the predictive uncertainty at points with low predicted mean, which causes sampling of EI and UCB to be too greedy. They therefore propose two new acquisition functions: Confidence Bound Minimization (CBM) and Expected Regret Minimization (ERM). Finally, the acquisition functions can only be used with tight bound but not with a more general bound, though the latter is more common. To solve this last issue, Nguyen et al. (2021) extend the model of Gunter et al. (2014) to a case where only a probability distribution for the lower bound of the objective function is known, and propose a new acquisition function called Bounded Entropy Search (BES). However, the other issues of linearly approximating the parabolic Gaussian processes remain.

Another candidate surrogate for the case of a known lower bound $f^b$ would be a non-negativity GP (Pensoneault et al., 2020). This method uses a GP as its model but introduces constraints to the hyperparameter tuning process, enforcing the probability of each $f(\mathbf{x})$ crossing the known bound to be below 5%. However, this method faces a couple of challenges. First, it is primarily suitable for low-dimensional problems due to its computational cost. Second, the range of the non-negativity GP remains $(-\infty, \infty)$, which means that there is still a mismatch with the desired range $[f^b, \infty)$ of feasible objective values.

The primary issue of the above-mentioned models is their inability to properly model that the function $f$ is known to take values in the range $[f^b, \infty)$. Jensen et al. (2013) proposed regression models using truncated distribution (GP-TG) or Beta distribution (GP-BE) that have the desired range but lack an analytical form for posterior inference. They use a Laplace approximation for the surrogate model. Additionally, Monte Carlo sampling would be needed for the acquisition function if this model were used in a BO framework. The method presented in this paper has a closed form and thus does not require costly approximations.

## 3 Bound-aware Bayesian optimization

In this section, we present the Bound-aware Bayesian optimizer (BABO) and its two key components: the surrogate model, *Shifted Logarithmic Gaussian Process* (SlogGP) that can leverage a lower bound $f^b$ about the global minimum $f^*$ if available, and the corresponding acquisition function *Shifted Logarithmic Truncated Expected Improvement* (SlogTEI).

### 3.1 The SlogGP Surrogate Model

The SlogGP model for an objective function $f(\cdot)$ is:

$$f(\mathbf{x}) = e^{g(\mathbf{x})} - \zeta,$$

where $g(\mathbf{x})$ is a GP and $\zeta$ is the *shift*, a parameter that is learned from data during model fitting. The mean of $f(\mathbf{x})$ is $e^{\mu(\mathbf{x}) + \frac{1}{2}\sigma^2(\mathbf{x})} - \zeta$ and its variance is $(e^{\sigma^2(\mathbf{x})} - 1)e^{2\mu(\mathbf{x}) + \sigma^2(\mathbf{x})}$, where $\mu(\mathbf{x})$ and $\sigma^2(\mathbf{x})$ are the mean and variance of $g(\mathbf{x})$.

This model can also be expressed as $\ln(f(\mathbf{x}) + \zeta) = g(\mathbf{x})$. When training the model, we set the mean of $g(\mathbf{x})$ to be the mean of the warped observed function values, i.e. the mean of $g(\mathbf{x})$ is $\frac{\sum_{i=1}^{N} \ln(y_i + \zeta)}{N}$ and $\mathbf{y} = [y_1, ...., y_N]^T$ is the observations.

The model can be viewed as a type of warped Gaussian process (Snelson et al., 2003), so we can learn the hyperparameters and parameters of the model, in particular the hyperparameters of the covariance function and shift $\zeta$ by minimizing the negative log likelihood with respect to $\mathbf{y} = [y_1, ...., y_N]^T$:

$$\mathcal{NLL} = \frac{1}{2}\ln(\det \mathbf{K}) + \frac{1}{2}W(\mathbf{y})^\top \mathbf{K}^{-1} W(\mathbf{y}) - \sum_{i=1}^{N} \ln\left(\frac{N-1}{N} \cdot \frac{1}{y_i + \zeta}\right) + \frac{N}{2}\ln(2\pi),$$

where $\mathbf{K}$ is the covariance matrix and $W(y) = \ln(y + \zeta) - \frac{\sum_{i=1}^{N} \ln(y_i + \zeta)}{N}$.

The SlogGP model $f(\mathbf{x}) = e^{g(\mathbf{x})} - \zeta$ can be viewed as combining aspects of a parabolic GP $f(\mathbf{x}) = \frac{1}{2}g^2(\mathbf{x}) - \zeta$ (Gunter et al., 2014; Ru et al., 2018; Nguyen & Osborne, 2020) and a log-transformed GP $f(\mathbf{x}) = e^{g(\mathbf{x})}$ (Hutter et al., 2009; 2011). By integrating these two methods, the SlogGP model offers several advantages. Specifically, in comparison to log-transformed GPs, SlogGPs can be utilized without requiring any lower bound information and exhibit greater expressiveness due to the learnable shift parameter $\zeta$. When compared to parabolic GPs, SlogGPs do not require any approximation and can guarantee an analytical form of the acquisition function (see Section 3.3). More importantly, as demonstrated in Section 3.2, SlogGPs possess greater expressiveness than standard GPs, given limited observation, a property that the other two models lack.

To the best of our knowledge, it is the first time that this model is proposed for handling known lower bound conditions. Additionally, we find that SlogGP-based BO outperforms GP-based BO even without any bound information, which we attribute to the enhanced expressiveness of SlogGPs.

### 3.2 Properties of the SlogGP Model

We begin with the case that we do *not* have a lower bound $f^* \geq f^b$ and show that a SlogGP is more general than a standard GP. In fact, as Theorem 3.1 shows, a SlogGP is reduced to a standard GP under certain conditions.

Note that a covariance function $K$ can be written as $K(\mathbf{x}_1, \mathbf{x}_2) = \sigma_g^2 \cdot k(\mathbf{x}_1, \mathbf{x}_2 | \theta_g)$, where $\sigma_g^2$ is a scaling hyperparameter called signal variance. $\theta_g$ are other hyperparameters that are independent of $\sigma_g$. For a noiseless GP, the covariance between $\mathbf{x}_1$ and $\mathbf{x}_2$ is $Cov(\mathbf{x}_1, \mathbf{x}_2) = K(\mathbf{x}_1, \mathbf{x}_2)$.

**Theorem 3.1.** *Define a SlogGP $f(\mathbf{x}) = e^{g(\mathbf{x}) + \mu_g} - \zeta$ with $g(\mathbf{x})$ a Gaussian process with zero mean, zero noise and a covariance function $K$ with signal variance $\sigma_g^2$ and other hyperparameters $\theta_g$.*

*Then for any GP $h(\mathbf{x})$ with mean $\tilde{\mu}_g$, signal variance $\tilde{\sigma}_g$, and other hyperparameters $\tilde{\theta}_g$, we can have the SlogGP $f(\mathbf{x})$ converge to $h(\mathbf{x})$ in distribution for any $\mathbf{x} \in \mathcal{X}$ by setting*

$$\begin{cases} \mu_g = \ln\left(\zeta + \tilde{\mu}_g\right) \\ \theta_g = \tilde{\theta}_g \\ \sigma_g = \dfrac{\tilde{\sigma}_g}{\zeta + \tilde{\mu}_g} \end{cases}$$

*and letting $\zeta \to \infty$, i.e.,*

$$\lim_{\zeta \to \infty} \mathcal{P}_{f(\mathbf{x})}(z) = \mathcal{P}_{h(\mathbf{x})}(z) \quad \forall z \in (-\zeta, \infty), \; \forall \mathbf{x} \in \mathcal{X},$$

*where $\mathcal{P}(\cdot)$ denotes the probability density function of a random variable.*

The proof for Theorem 3.1 can be found in Appendix A.4.

Hence, in the limit, a GP is a special case of a SlogGP. Under the conditions specified in Theorem 3.1, as $\zeta \to \infty$, the lower bound $-\zeta$ tends to negative infinity and the skewness converges to zero, causing the distribution to converge to a Gaussian distribution. This relationship is unidirectional: while SlogGP can approximate a GP by setting its skewness parameter near zero, a symmetric GP inherently lacks the flexibility to approximate a skewed SlogGP distribution.

Figure 1 illustrates Theorem 3.1 by showing 200 samples from SlogGPs and GPs with three different covariance functions: RBF, Matern32 and Brownian. The left column contains samples from GPs. The middle column contains samples from SlogGPs whose parameters satisfy Theorem 3.1 (considering $\zeta \to \infty$ cannot be achieved numerically, we set $\zeta$ to be a large number $\zeta := 100$) to approximate the GPs on the left side. The right column contains SlogGPs with $\zeta = 0.5$. We can see that the posterior distributions of the left and the middle are very close. If the shift $\zeta$ is not that large, we find that the distribution is more skewed, as is shown in the right side.

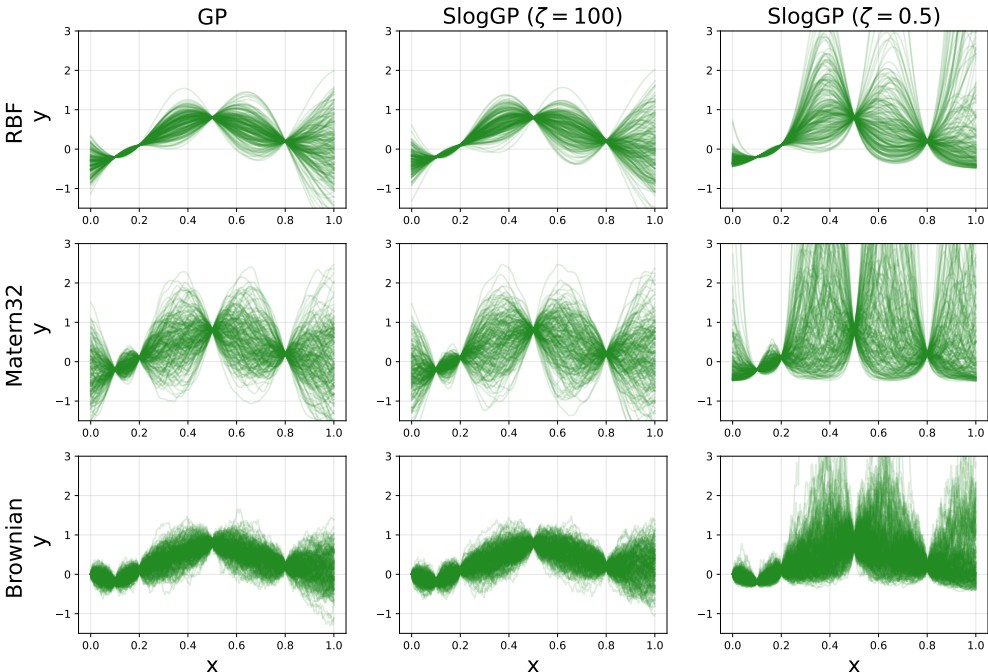

Figure 1: Example of surrogate models (1D) for (left) a normal GP (center) a SlogGP with the shift $\zeta := 100$ and parameters set according to Theorem 3.1 to match the normal GP, and (right) a SlogGP with the shift $\zeta := 0.5$ .

### 3.3 An adaptation of EI to the SlogGP Model

Expected improvement (EI) is a popular acquisition function. Recall that for a GP model, the posterior predictive distribution of function values at any particular location $\mathbf{x}$ is Gaussian and thus EI has a simple analytical form. However, for SlogGP, the posterior predictive distribution is no longer Gaussian and hence we have to adapt EI.

We derive a closed-form expression for Expected Improvement under the proposed surrogate model, which we term Shifted Logarithmic Expected Improvement (SlogEI):

$$\alpha^{\text{SlogEI}}(\mathbf{x}; f_{min}) = (f_{min} + \zeta) \cdot \Phi\left(\frac{\ln(f_{min} + \zeta) - \mu(\mathbf{x})}{\sigma(\mathbf{x})}\right)$$
$$- e^{\mu(\mathbf{x}) + \frac{\sigma^2(\mathbf{x})}{2}} \cdot \Phi\left(\frac{\ln(f_{min} + \zeta) - \mu(\mathbf{x}) - \sigma^2(\mathbf{x})}{\sigma(\mathbf{x})}\right).$$

*Derivation* 1. For a solution $\mathbf{x}$ and the current minimal observation $f_{min}$, we denote the predicted mean of $g(\mathbf{x})$ as $\mu(\mathbf{x})$ and variance of $g(\mathbf{x})$ as $\sigma^2(\mathbf{x})$. Due to the normal distribution of $g(\mathbf{x})$, $e^{g(\mathbf{x})}$ (denoted by $Z$) follows a log-normal distribution.

To calculate SlogEI, we firstly calculate $\mathbb{E}[(\eta - Z)^+]$ using the definition of the log-normal distribution, where $\eta$ is a constant and $(\cdot)^+$ denotes the positive part function:

$$\mathbb{E}[(\eta - Z)^+] = \int_0^\infty (\eta - Z)^+ \cdot \frac{1}{z\sigma(\mathbf{x})\sqrt{2\pi}} e^{-\frac{(\ln(z) - \mu(\mathbf{x}))^2}{2\sigma(\mathbf{x})^2}} dz$$
$$= \int_0^\eta (\eta - z) \cdot \frac{1}{z\sigma(\mathbf{x})\sqrt{2\pi}} e^{-\frac{(\ln(z) - \mu(\mathbf{x}))^2}{2\sigma(\mathbf{x})^2}} dz + \int_\eta^\infty 0 \cdot \frac{1}{z\sigma(\mathbf{x})\sqrt{2\pi}} e^{-\frac{(\ln(z) - \mu(\mathbf{x}))^2}{2\sigma(\mathbf{x})^2}} dz$$
$$= \int_0^\eta (\eta - z) \cdot \frac{1}{z\sigma(\mathbf{x})\sqrt{2\pi}} e^{-\frac{(\ln(z) - \mu(\mathbf{x}))^2}{2\sigma^2(\mathbf{x})}} dz$$
$$= \eta \cdot \int_0^\eta \frac{1}{z\sigma(\mathbf{x})\sqrt{2\pi}} e^{-\frac{(\ln(z) - \mu(\mathbf{x}))^2}{2\sigma(\mathbf{x})^2}} dz - \int_0^\eta z \cdot \frac{1}{z\sigma(\mathbf{x})\sqrt{2\pi}} e^{-\frac{(\ln(z) - \mu(\mathbf{x}))^2}{2\sigma(\mathbf{x})^2}} dz$$
$$= \eta\Phi\left(\frac{\ln(\eta) - \mu(\mathbf{x})}{\sigma(\mathbf{x})}\right) - \int_0^\eta z \cdot \frac{1}{z\sigma(\mathbf{x})\sqrt{2\pi}} e^{-\frac{(\ln(z) - \mu(\mathbf{x}))^2}{2\sigma(\mathbf{x})^2}} dz$$

The last equality simply uses the definition of the CDF of the log-normal distribution. The second term is the partial expectation, so

$$\int_0^\eta z \cdot \frac{1}{z\sigma(\mathbf{x})\sqrt{2\pi}} e^{-\frac{(\ln(z) - \mu(\mathbf{x}))^2}{2\sigma^2}} dz$$
$$= e^{\mu(\mathbf{x}) + \frac{\sigma(\mathbf{x})^2}{2}} \cdot \Phi\left(\frac{\ln(\eta) - \mu(\mathbf{x}) - \sigma(\mathbf{x})^2}{\sigma(\mathbf{x})}\right)$$

Hence,

$$\mathbb{E}[(\eta - Z)^+] = \eta\Phi\left(\frac{\ln(\eta) - \mu(\mathbf{x})}{\sigma(\mathbf{x})}\right) - e^{\mu(\mathbf{x}) + \frac{\sigma(\mathbf{x})^2}{2}} \cdot \Phi\left(\frac{\ln(\eta) - \mu(\mathbf{x}) - \sigma(\mathbf{x})^2}{\sigma(\mathbf{x})}\right)$$

Setting $\eta = f_{min} + \zeta$, we obtain the formula of $\mathbb{E}[(f_{min} - (Z - \zeta))^+]$, i.e. SlogEI:

$$\alpha^{\text{SlogEI}}(\mathbf{x}; f_{min}) = (f_{min} + \zeta) \cdot \Phi\left(\frac{\ln(f_{min} + \zeta) - \mu(\mathbf{x})}{\sigma(\mathbf{x})}\right)$$
$$- e^{\mu(\mathbf{x}) + \frac{\sigma^2(\mathbf{x})}{2}} \cdot \Phi\left(\frac{\ln(f_{min} + \zeta) - \mu(\mathbf{x}) - \sigma^2(\mathbf{x})}{\sigma(\mathbf{x})}\right).$$

$\square$

A visualization comparing SlogGP+SlogEI (Figure 17) with GP+EI (Figure 16) can be found in Appendix A.3.

Note that for $\zeta := 0$, SlogEI becomes the acquisition function by Hutter et al. (2011). Furthermore, when the posterior mean and variance of $g(\mathbf{x})$ are differentiable with respect to $\mathbf{x}$, the SlogEI acquisition function inherits this differentiability. The corresponding gradient calculations are also provided in Appendix A.2, along with an adaptation of the popular Probability of Improvement acquisition criterion.

In Figure 2, we compare GP+EI and SlogGP+SlogEI empirically on objective functions that are drawn from a GP model, shown in the left plot, and a SlogGP model as shown in the right one. The hyperparameters of both methods are fit to data. We observe that both methods perform similarly when the objective functions are sampled from a GP. When $f$ is sampled from a SlogGP, the performance of GP+EI degrades substantially. This observation is consistent with the theoretical property shown above that a SlogGP model can approximate a GP sample, but not vice versa. We also performed a cross-validation test on the prediction error of the two surrogate models, see Table 1. When the objective function is a GP, SlogGP and GP are equally good in prediction while when the objective function is a SlogGP, SlogGP achieves a better cross-validation error. Please see Appendix A.5 for details about the experimental setup.

Table 1: Cross-validation test. The column determines the test function and the row the choice of the surrogate model. The cells give the prediction error of the surrogate.

|  | GP | SlogGP |
|---|---|---|
| GP | $0.828(\pm0.0801)$ | $6.21(\pm1.16)$ |
| SlogGP | $0.841(\pm0.0802)$ | $1.414(\pm0.287)$ |

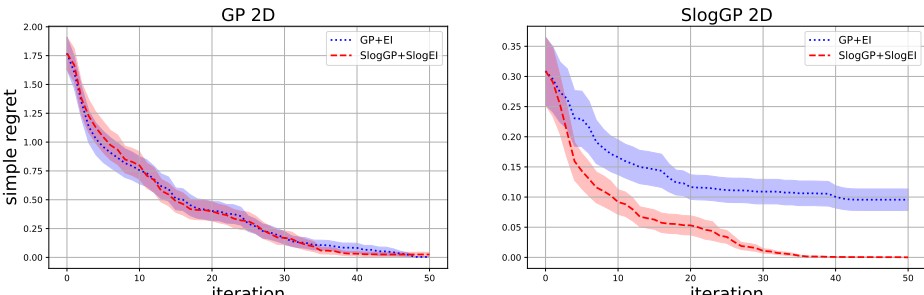

Figure 2: Within-model Test. The left panel shows the performance of GP+EI and SlogGP+SlogEI on GP generated functions, while the right panel compares the two algorithms on SlogGP generated functions. While both models work equally well on GP generated functions, only the SlogGP model works well on the SlogGP generated functions.

Next we show how the lower bound information $f^* \geq f^b$ can be incorporated into the SlogGP surrogate model. Then we present a novel acquisition function called SlogTEI that also leverages the lower bound $f^b$.

### 3.4 Incorporating a lower bound on the minimum into SlogGP

An advantage of the SlogGP model is that it allows to enforce a lower bound through its parameter $\zeta$. On the other hand, the skewness introduced may not be a very good fit to the observed data, creating a possible conflict between setting $\zeta$ to a value that best represents the bound information, and setting $\zeta$ to a value that best matches the observed data.

To solve this potential conflict, we use a maximum a posteriori (MAP) estimate for $\zeta$ and include the bound information only as prior. Furthermore, we implemented some mechanisms that allow to quickly reduce the reliance on the prior if the observed mismatch between prior and fit to the data is very large. The details of how we choose $\zeta$ are explained in the following.

When the lower bound $f^b$ is known, a straightforward way is to set $\zeta = -f^b$. However, this method may suffer if the prior information is not accurate enough or would otherwise lead to a model mismatch. For instance, suppose that the objective function is SlogGP $f(\mathbf{x}) = e^{g(\mathbf{x})} + 10$, and the prior information is $f(\mathbf{x}) > 0$. In this case, the prior information is correct, but if we enforce $\zeta = 0$, the bound is too loose and the model skewness doesn't match the skewness of the true function. Thus, any information about the lower bound should only guide, but not dictate $\zeta$. A natural way is to use a maximum a posteriori probability (MAP) estimate for the parameter $\zeta$. We choose the prior distribution to be a shifted log-normal distribution to guarantee that $-\zeta < f_{min}$, where $f_{min}$ is the current best value and $-\zeta$ is the lower bound of the model, so that the model is well-defined:

$$\zeta \sim -f_{min} + e^Z, \tag{1}$$

where $Z \sim \mathcal{N}(\ln(f_{min} - f^b), 2\ln(f_{min} - f^b + \delta_1) - 2\ln(f_{min} - f^b))$, and $\delta_1$ is a positive hyperparameter, representing the gap between the mean and median of the prior distribution of $\zeta$, which implicitly determines its variance. We set the mean and variance of $Z$ such that the median of $-\zeta$ equals the known lower bound $f^b$ and the mean of $-\zeta$ equals $f^b - \delta_1$. The approach is not sensitive to the selection of $\delta_1$, requiring only a small positive value. In our experiments, we set $\delta_1 = 0.1$.

Sometimes, there can be a prior-data conflict, where the prior information is inconsistent with observation data. Specifically, in SlogGP, a prior-data conflict is detected when the estimated lower bound $-\hat{\zeta}$ is unlikely under the prior distribution. Note that in this paper, we distinguish $\zeta$ and $\hat{\zeta}$: $\zeta$ is an unknown parameter while $\hat{\zeta}$ is its estimator. To handle a prior-data conflict, we introduce an uncertainty level and variance threshold. When we observe the MAP estimator $-\hat{\zeta}$ is unlikely under the prior distribution, we reduce our confidence through uncertainty level $U$. Specifically, we control the variance of $Z$ to be $U^2 \cdot 2(\ln(f_{min} - f^b + \delta_1) - \ln(f_{min} - f^b))$ so that as the uncertainty level $U$ increases, the prior bound information becomes weaker. Thus, we set $Z$ in the prior in Eq. 1 to be $Z \sim \mathcal{N}(\ln(f_{min} - f^b), U^2(2\ln(f_{min} - f^b + \delta_1) - 2\ln(f_{min} - f^b)))$.

Initially, the uncertainty level is set to be $U = 1$. When $\hat{\zeta}$ lies in the tails of the prior distribution, specifically when $F_{\zeta_{prior}}(\hat{\zeta}) < \delta_2$ or $F_{\zeta_{prior}}(\hat{\zeta}) > 1 - \delta_2$ (where $F_{\zeta_{prior}}$ is the cumulative density function of $\zeta_{prior}$ and $\delta_2$ is a hyperparameter that sets the threshold for detecting prior conflicts), then we interpret this as a conflict between the prior information and the observed data, and re-train the surrogate model by maximum likelihood estimation (MLE) as the MAP depends on the prior information that we now consider unreliable. In the next iteration, we increase the uncertainty level $U$ by multiplying it with the absolute value of standard score of the estimated $\hat{\zeta}$, which can be calculated as $\left| \frac{\ln(\hat{\zeta} + f_{min}) - \mu_{prior}}{\sigma_{prior}} \right|$ for the shifted log-normal distribution (where $\mu_{prior}$ and $\sigma_{prior}^2$ refer to the mean and the variance of $Z$ in the prior distribution). A large standard score absolute value indicates a prior-data mismatch, prompting us to reduce reliance on the prior by increasing the variance in subsequent steps. $\delta_2$ needs to be a small probability. In this paper, we set $\delta_2 = 0.01$.

Additionally, we will exclude prior information when the estimated lower bound $-\hat{\zeta}$ significantly deviates from the known lower bound $f^b$ (i.e., when $|-\hat{\zeta} - f^b|$ exceeds a threshold). However, setting a universal threshold is challenging due to the varying ranges of objective functions. Instead, we leverage Theorem 3.1, which establishes that $\sigma_g = \hat{\sigma}_g/(\zeta + \hat{\mu}_g) \to 0$ as $\zeta \to \infty$. Consequently, we use the estimated signal variance $\sigma_g^2$ as an indicator for incorporating bound information. Specifically, to prevent prior-data conflict, we disregard bound information when $\sigma_g^2 < \delta_3$, where $\delta_3$ is a hyperparameter that determines when SlogGP's behavior approximates a GP closely enough. We set $\delta_3 = 0.25^2$ in our experiments.

As we show in Section 5.2, our algorithm is not sensitive to the choice of $\delta_1$, $\delta_2$ and $\delta_3$.

Figure 3 illustrates a GP model, a SlogGP model and a SlogGP model with the prior knowledge of $f^*$.

## 3.5 Incorporating knowledge of $f^* \geq f^b$ into the acquisition criterion

A potential drawback of SlogEI is that the improvement is calculated over the range $(-\hat{\zeta}, f_{min}]$, where the estimate $-\hat{\zeta}$ can be smaller than the known lower bound $f^b$. Thus, the Shifted Logarithmic Truncated Expected Improvement (SlogTEI) acquisition criterion truncates any impossible value below $f^b$ when calculating SlogEI:

$$\alpha^{\text{SlogTEI}}(\mathbf{x}; f_{min}, f^b) = \alpha^{\text{SlogEI}}(\mathbf{x}; f_{min}) - \alpha^{\text{SlogEI}}(\mathbf{x}; f^b).$$

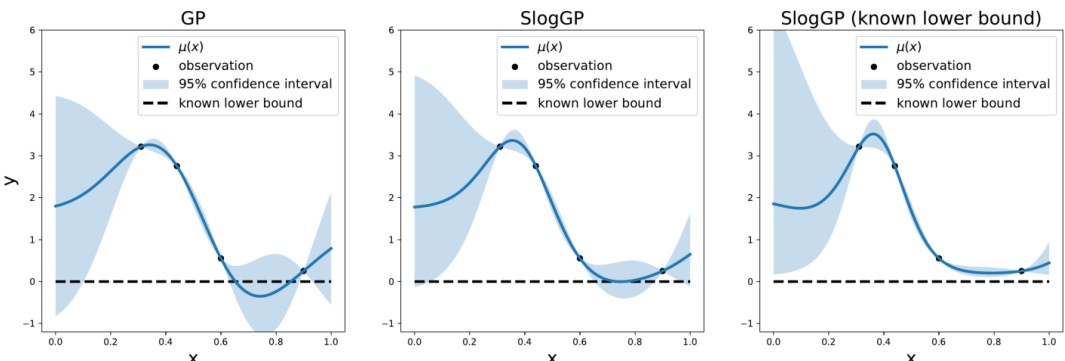

Figure 3: The predictive posterior distribution of the SlogGP model respects the known lower bound. (left) shows a GP surrogate model; (middle) shows a SlogGP surrogate model; (right) shows a SlogGP surrogate model with known lower bound.

Note that if the estimate $-\hat{\zeta}$ is larger than the known lower bound, i.e $-\hat{\zeta} > f^b$, $\alpha^{\text{SlogTEI}}(\mathbf{x})$ becomes $\alpha^{\text{SlogEI}}(\mathbf{x})$. Additionally, SlogTEI is differentiable as it is the difference of two differentiable acquisition functions.

We summarize the BABO method that uses the SlogGP$^b$ surrogate and the SlogTEI acquisition criterion in Algorithm 1, where the superscript $b$ means that the SlogGP is trained with lower bound information.

---

**Algorithm 1** BABO (SlogGP$^b$ + SlogTEI)

---

1: Initial data points $\mathcal{D}_0$ and uncertainty level $U = 1$
2: Known lower bound $f^b$
3: **for** $n = 0$ **to** $T$ **do**
4:    Set prior distribution $\zeta_{prior}$: Eq. 1
5:    Train a surrogate model $\hat{f}(\cdot)$ by MAP with $\mathcal{D}_n$ and prior distribution $\zeta_{prior}$
6:    **if** $F_{\zeta_{prior}}(\hat{\zeta}) < \delta_2$ or $F_{\zeta_{prior}}(\hat{\zeta}) > 1 - \delta_2$ **then**
7:       Train a surrogate model $\hat{f}(\cdot)$ by MLE with $\mathcal{D}_n$
8:       $U = U \cdot \left| \frac{\ln(\hat{\zeta}+f_{min}) - \mu_{prior}}{\sigma_{prior}} \right|$
9:    **end if**
10:   **if** $\hat{f}$ is trained by MAP and $\hat{\sigma}_g^2 < \delta_3$ **then**
11:      Train a surrogate model $\hat{f}(\cdot)$ by MLE with $\mathcal{D}_n$
12:   **end if**
13:   Find $\mathbf{x}_{n+1} = \arg\max_{\mathbf{x} \in \mathcal{X}} \{ \alpha^{\text{SlogTEI}}(\mathbf{x}) \}$
14:   Evaluate the objective function $y_{n+1} = f(\mathbf{x}_{n+1})$
15:   Update $\mathcal{D}_{n+1} = \mathcal{D}_n \cup \{ \mathbf{x}_{n+1}, y_{n+1} \}$
16: **end for**

---

Note that BABO can be straightforwardly extended to batch acquisition by adapting the qEI idea from Ginsbourger et al. (2008). The acquisition function value $\alpha^{\text{qSlogTEI}}(X; f_{min}, f^b)$ for a batch of solutions $X = (\mathbf{x}_1, \dots, \mathbf{x}_q)$ is calculated by Monte Carlo simulation: $\alpha^{\text{qSlogTEI}}(X; f_{min}, f^b) = \frac{1}{N_{MC}} \sum_{i=1}^{N} \max_{j=1,\dots,q} \left\{ (f_{min} - \xi_{ij})^+ - (f^b - \xi_{ij})^+ \right\}$, where $(\cdot)^+$ denotes the positive part function, $N_{MC}$ is the number of samples and $\xi_i \sim \mathbb{P}(f(X) \mid \mathcal{D})$.

Similarly, we propose Truncated Expected Improvement (TEI) for vanilla GPs, an adaptation of Expected Improvement (EI) that uses a lower bound $f^b$ on the global minimum. When we know that the lower bound

is $f^b$, we truncate the distribution of $f(\mathbf{x})$ at $f^b$ and calculate TEI as

$$\alpha^{\text{TEI}}(\mathbf{x}; f_{min}, f^b) = \int_{f^b}^{f_{min}} (f_{min} - z) \cdot \phi \left( \frac{z - \mu(\mathbf{x})}{\sigma(\mathbf{x})} \right) dz$$
$$= \mathbb{E}[(f_{min} - f(\mathbf{x}))^+] - \mathbb{E}[(f^b - f(\mathbf{x}))^+]$$

TEI will serve as a benchmark algorithm in our experiments.

## 4 Experiments

In this section, we compare performance of BABO with other benchmark algorithms across various test functions. The competing methods are:

- Random (Bergstra et al., 2011): randomly choosing the next solution.

- EI (Mockus, 1998): using Expected Improvement (EI) as the acquisition function

- TEI: using Truncated EI, see Section 3.5

- MES$^b$ (Wang et al., 2018): adapted max-entropy search, see Section 2.2

- OBCGP (Jeong & Kim, 2021): OBCGP follows the paper by Jeong & Kim (2021). The original implementation of Jeong & Kim (2021) lacks version information and depends on deprecated packages no longer available through pip. We have updated the codebase to utilize current package versions and fixed existing bugs.

- ERM (Nguyen & Osborne, 2020): Our parabolic GP+ERM algorithm follows the method of Nguyen & Osborne (2020) and the code shared by the authors. Initially, we employ the regular GP+EI until the lower confidence bound (LCB) reaches the known lower bound. Subsequently, we switch to the transformed GP+ERM algorithm. This ensures a seamless integration between the two methods, allowing for an effective exploration-exploitation trade-off during the optimization process. In addition, following their code, if the next evaluation $\mathbf{x}_{n+1}$ is too close to an existing observation (1-norm distance smaller than $3d \cdot 10^{-4}$), we will pick a random $\mathbf{x}$ instead. The hyperparameter $\beta$ of LCB is set to be $\sqrt{\ln(N)}$.

We also fix $-\zeta = f^b$ in BABO to understand the benefit of learning $\zeta$. Note that when $-\zeta = f^b$, SlogTEI will become equivalent to SlogEI, and BABO with fixed $\zeta$ can be viewed as an extension of the method in Hutter et al. (2011) where in a preprocessing step the function is shifted into the positive region based on the information of the lower bound. We evaluate the algorithms on eight synthetic functions that are widely used in BO testing as well as three real-world problems. In addition, we compare qBABO with qEI on two synthetic functions with different batch sizes and the results are shown in Figure 18 in Appendix A.6. The code is available at `https://github.com/HanyangHenry-Wang/BABO.git`.

Following the experimental setting in Nguyen & Osborne (2020); Jeong & Kim (2021), we assume noise-free observations in experiments. To ensure numerical stability, we introduce a small, adaptive noise variance proportional to the estimated signal variance (more details can be found in Appendix A.5). Additionally, as discussed in Section 3, the hyperparameters of BABO are set to be $(\delta_1, \delta_2, \delta_3) = (0.1, 0.01, 0.25^2)$. We will later show in Section 5.2 that BABO's performance is robust to hyperparameter values. Plots show the simple regret (mean $\pm$ one standard error) over 100 repetitions for all functions except 50 repetitions for the 10d function and 20 repetitions for the PDE Variance problem. For improved visualization, some results are presented using logarithmic scales on the $y$-axis, as indicated by their $y$-axis scale. Appendix A.5 gives more details of the experimental setup.

## 4.1 Synthetic Test Functions

We compare the performances on eight synthetic functions with 2 to 10 dimensions where we have an exact bound on the minimum objective value. The known lower bound is thus set to the minimal value of the objective function, i.e., $f^b := f^*$, which is also the setting for which ERM was proposed.

Figure 4 summarizes the performances. Dashed lines indicate algorithms that do not use any information about $f^*$, whereas solid lines correspond to methods that do leverage bound information. We observe that BABO (black line) usually achieves substantially better objective values at the same iteration than the other algorithms. An exception is the Ackley (6D) benchmark where OBCGP achieves better values and BABO comes in second.

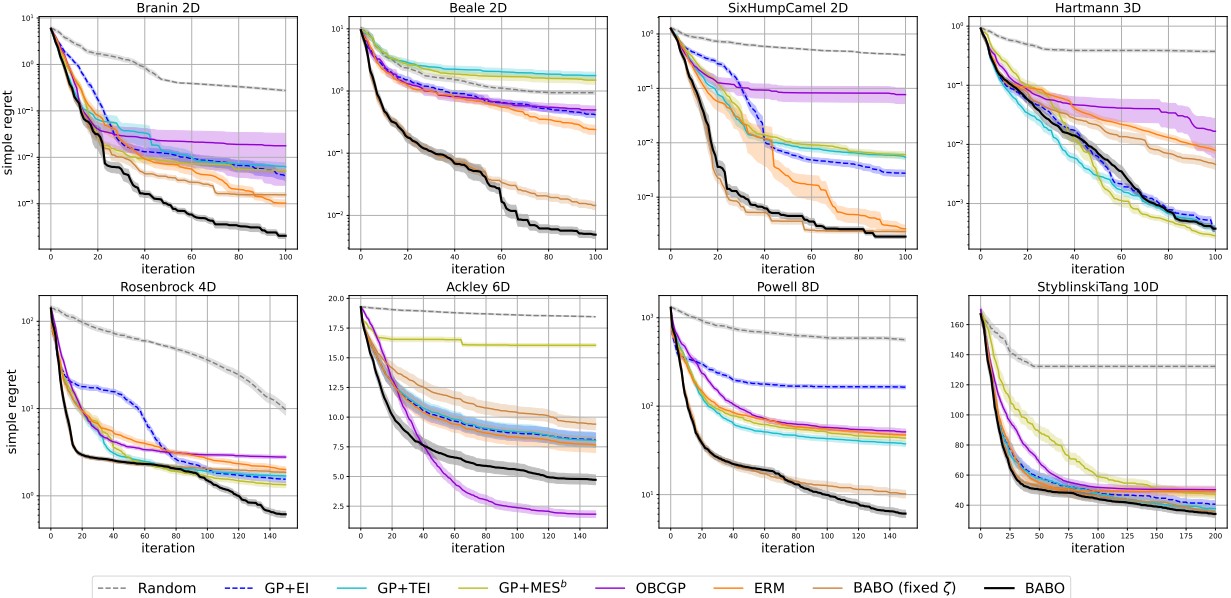

Figure 4: Experiment Results (Synthetic Functions): We observe that BABO works best except in Ackley (6D), where OBCGP works better.

## 4.2 XGBoost Hyperparameter Tunning Task

In the XGBoost Hyperparameter Tunning Task, the goal is to optimize six hyperparameters of an XGBoost model (Chen & Guestrin, 2016) for various classification problems. We chose four: Skin Segmentation, BankNote Authentication, Wine Quality, and Breast Cancer. The six hyperparameters are min child weight, colsample bytree, max depth, subsample, alpha, and gamma. Notably, max depth is integer-valued; however, we conduct the search in a continuous space and round it to the nearest integer for evaluation. The objective value is determined by the model's classification error rate on a hold-out dataset. Since we aim to minimize the error rate in BABO, the objective function has a natural lower bound of 0%. However, for visualization purposes, we present the results in terms of accuracy, as this is the standard metric reported in classification problems.

Figure 5 shows that BABO achieves the best performance in both the Skin Segmentation and Breast Cancer tasks. In the BankNote Authentication task, BABO and BABO (fixed $\zeta$) demonstrate nearly identical performance, with BABO (fixed $\zeta$) showing a marginally higher mean at the final iteration. In the Wine Quality task, OBCGP exhibits a slight advantage over BABO. Overall, BABO demonstrates consistent and superior performance compared to competitor algorithms.

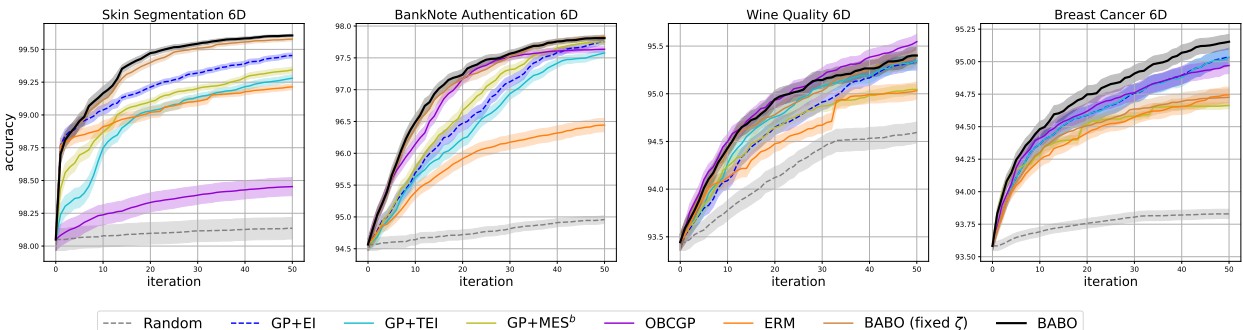

Figure 5: An empirical evaluation of BABO and competitors on XGBoost hyperparameter tuning tasks. BABO achieves overall best performance.

## 4.3 PDE Variance Problem

The PDE Variance Problem (Maddox et al., 2021) aims to minimize the output variance of a spatially-coupled Brusselator system by optimizing four continuous parameters that control its diffusivity and reaction rates. This problem is also used as a benchmark problem in Li et al. (2024). The lower bound is zero as variance is non-negative. Since each function evaluation requires computationally expensive simulation, we limit our experiments to 20 repetitions.

Figure 6(a) shows that BABO converges more quickly than other algorithms, followed by BABO (fixed $\zeta$).

## 4.4 The Robot Push Problem

The goal of the $4D$ Robot Push problem (Wang & Jegelka, 2017; De Ath et al., 2021) is to direct a robot to push a ball towards an unknown target, minimizing the distance between the ball and the target. The distance is nonnegative and thus implies a lower bound of zero for $f^*$. The first input variables determine the initial location of the robot, the third determines the angle of its rectangular hand, and the fourth sets the number of time-steps that the robot is to move.

Figure 6(b) shows that BABO achieves better solutions at the same iteration and converges more quickly. The runner-up is TEI that also uses our truncated EI acquisition function but a regular GP surrogate model.

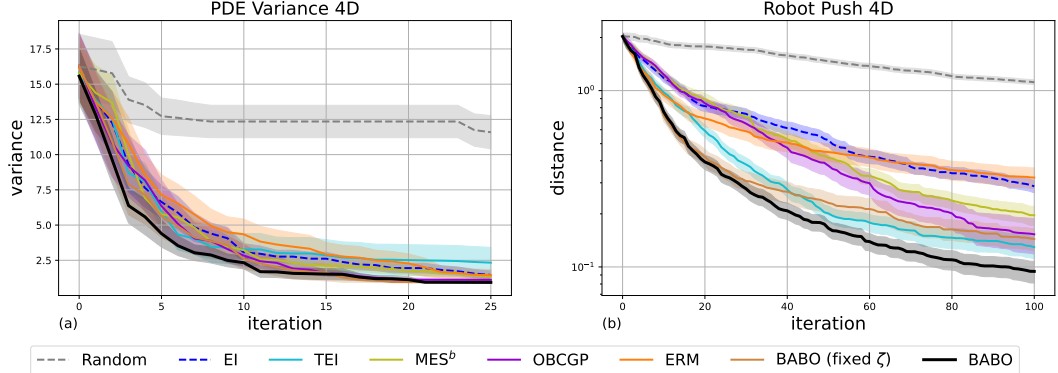

Figure 6: An empirical evaluation of BABO and competitors on the PDE Variance problem and the Robot Push problem. BABO achieves better solutions at the same iteration and converges more quickly.

To summarize the experimental results, we ranked the algorithms based on their mean at the final iteration, as shown in Table 2.

Table 2: Performance Rank

| Problem | BABO | Fixed $\zeta$ | TEI | EI | MES$^b$ | ERM | OBCGP | Random |
|---|---|---|---|---|---|---|---|---|
| Branin | **1** | 3 | 6 | 4 | 5 | 2 | 7 | 8 |
| Beale | **1** | 2 | 8 | 4 | 7 | 3 | 5 | 6 |
| SixHumpCamel | **1** | 2 | 5 | 4 | 6 | 3 | 7 | 8 |
| Hartmann | 2 | 5 | 3 | 4 | **1** | 6 | 7 | 8 |
| Rosenbrock | **1** | 5 | 4 | 3 | 2 | 6 | 7 | 8 |
| Ackley | 2 | 6 | 4 | 5 | 7 | 3 | **1** | 8 |
| Powell | **1** | 2 | 3 | 7 | 4 | 5 | 6 | 8 |
| StyblinskiTang | **1** | 2 | 3 | 4 | 5 | 6 | 7 | 8 |
| Skin | **1** | 2 | 5 | 3 | 4 | 6 | 8 | 8 |
| Bank | 2 | **1** | 6 | 4 | 3 | 7 | 5 | 8 |
| Wine | 2 | 3 | 4 | 5 | 6 | 7 | **1** | 8 |
| Breast | **1** | 6 | 2 | 3 | 7 | 5 | 4 | 8 |
| PDE | **1** | 2 | 7 | 6 | 4 | 5 | 3 | 8 |
| Robert | **1** | 3 | 2 | 6 | 5 | 7 | 4 | 8 |
| Average | **1.3** | 3.1 | 4.4 | 4.4 | 4.7 | 5.0 | 5.1 | 7.9 |

## 5  Investigating the influence of prior information on the optimum

In this section, we study the influence of prior information about $f^*$. First, we conduct an ablation study and observe that leveraging prior information about the optimum in model training and acquisition function is beneficial. Second, we conduct a hyperparameter sensitivity analysis to demonstrate BABO's robustness across different values of $\delta_1$, $\delta_2$, and $\delta_3$. Then, we investigate why we need the uncertainty level and variance threshold when using bound information in model training. Finally, we investigate how SlogGP$^b$ is influenced by its estimated lower bound $-\hat{\zeta}$ and lower bound information.

### 5.1  The Contributions of Different Components

To better understand how different components of BABO contribute to its success, we compare the following settings:

- SlogGP+SlogEI: Removing the bound information usage in both SlogGP and acquisition function. This will tell us how helpful the lower bound information really is.

- SlogGP$^b$+SlogEI: Removing bound information from the acquisition function only. This will show the relative importance of using bound information in the SlogGP model and the acquisition function.

- SlogGP+SlogTEI: Removing bound information from the model only. This will show the relative importance of using bound information in the SlogGP model and the acquisition function.

Looking at the results on the eight synthetic test functions in Figure 7 and two real-life problems in Figure 8, the full BABO performs best, often followed closely by the version using the bound information only in the SlogGP model or the version using the bound information only in the acquisition function, and then the version not using bound information at all. The standard GP+EI usually did quite poorly, except for the Hartmann function where all methods share similar performance. An additional observation is that SlogGP+SlogEI (not using bound information) works better than GP+EI, which can be explained by SlogGP's larger expressive power, as shown in Theorem 3.1. Also, it is worth noting that for the Ackley function, SlogGP+SlogEI is expected to perform similarly to GP+EI, as the learned signal variance $\hat{\sigma}$ is nearly zero. However, in our experiments, SlogGP+SlogEI outperforms GP+EI. This may be due to numerical issues. When optimizing the Ackley function, SlogGP learns a very small signal variance (approximately $10^{-5}$), and the resulting noise variance of $10^{-10}$. However, we observe that the effective noise (due to numerical issues) ends up to be around $10^{-6}$. This seems beneficial for the Ackley function, as it helps to handle Ackley's numerous local optima. This phenomenon is specific to Ackley due to the very small learned signal variance.

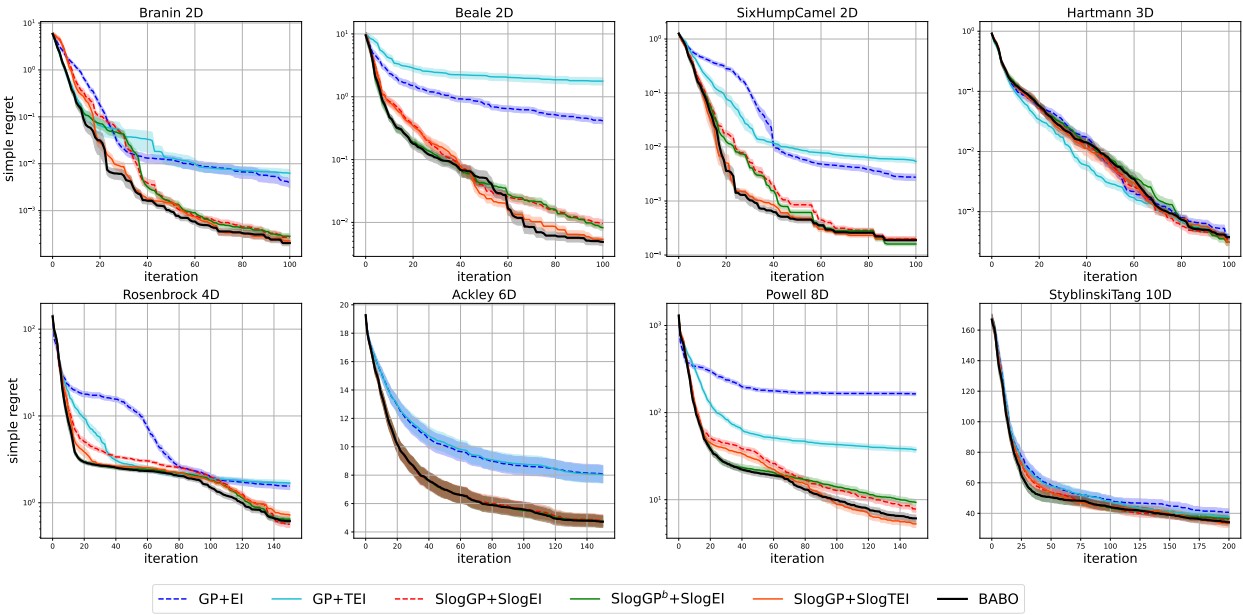

Figure 7: Different Components Comparison (Synthetic Functions): Generally, the complete BABO approach demonstrates superior performance, suggesting that incorporating bound information in both the modeling and acquisition function stages is beneficial.

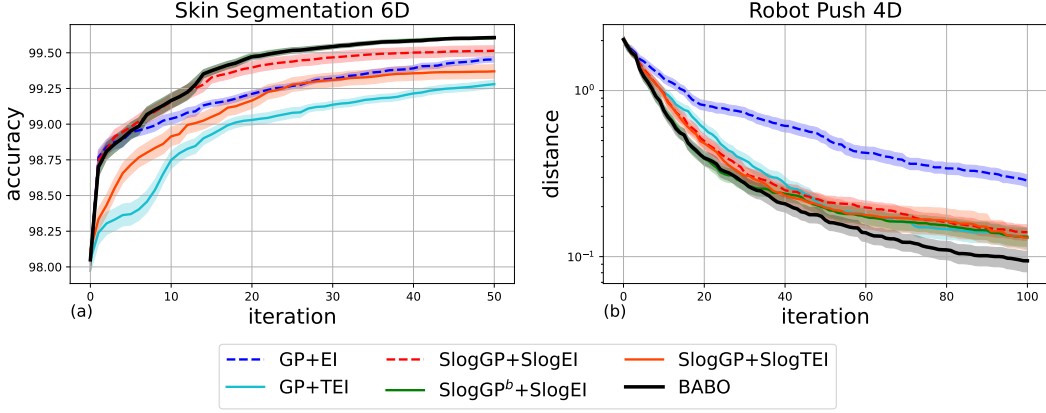

Figure 8: Different Components Comparison (Real-world Benchmarks): Generally, the complete BABO approach demonstrates superior performance, suggesting that incorporating bound information in both the modeling and acquisition function stages is beneficial.

## 5.2 Hyperparameter Sensitivity Analysis

BABO has three hyperparameters: $\delta_1$, $\delta_2$, and $\delta_3$. $\delta_1$ represents the gap between the mean and median of the prior distribution, which implicitly determines the variance of the prior distribution of $\zeta$. $\delta_2$ sets the threshold for detecting prior conflicts, while $\delta_3$ determines when SlogGP's behavior approximates a GP closely enough that we discard the prior information at the current iteration.

Figure 9 and Figure 10 demonstrate BABO's performance given different hyperparameter values across different objective functions. The Ackley function is a special case where the $\delta_3$ condition is consistently triggered, resulting in training without any bound information. Consequently, $\delta_1$ and $\delta_2$ do not influence its performance, so we only analyze its sensitivity to $\delta_3$. The experiments demonstrate that BABO exhibits

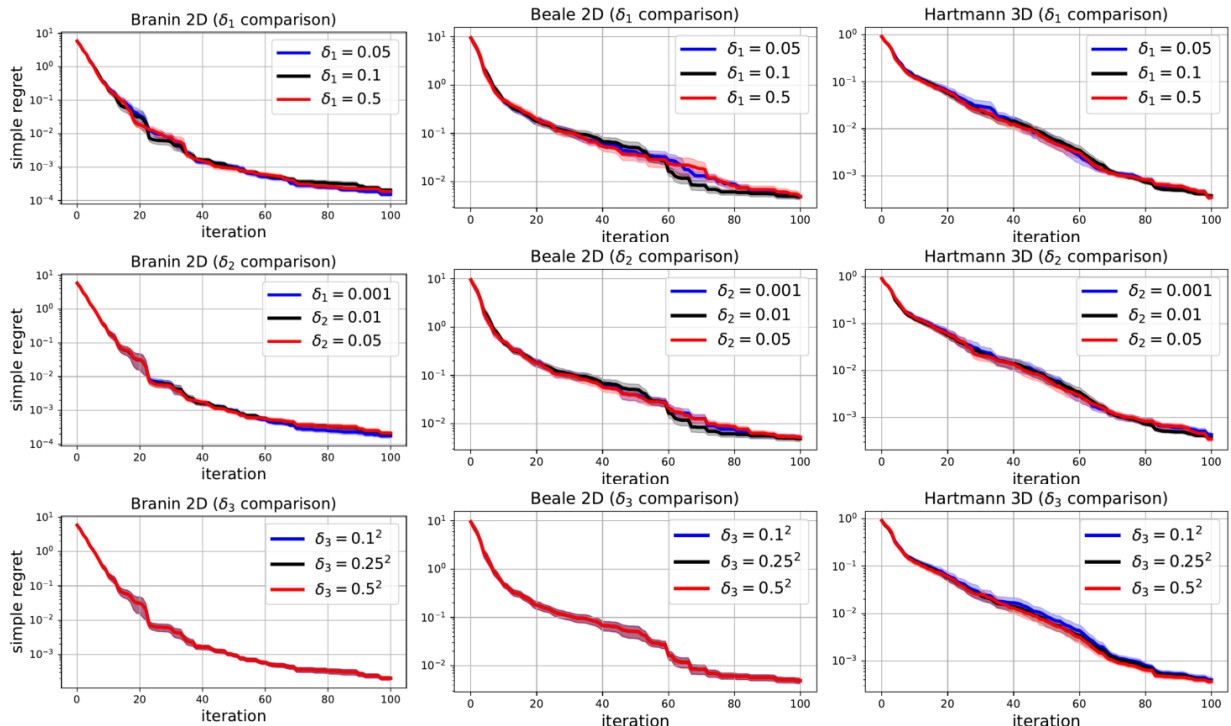

Figure 9: Hyperparameter sensitivity analysis shows that BABO is robust to different hyperparameter values except too small $\delta_3$ may lead to marginally degraded performance on the Hartmann function.

robust performance across different hyperparameter values, with only marginally degraded performance observed when $\delta_3$ is set too low.

This robustness can be explained as follows: Although $\delta_1$ determines the prior variance, which typically plays a crucial role in model training, our prior-conflict detection adjusts the prior variance when the data deviates from the prior bounds. For $\delta_2$, we observe that as the number of observed data points increases, if the bound information does not align with the data, the probability that $f^b$ serves as valid prior information will quickly drop and reach $\delta_2$, even when $\delta_2$ is relatively small. Hence, even when $\delta_2$ is set small, it can still detect prior-data conflict. For $\delta_3$, the choice does not matter for functions that can be modeled by SlogGP well (such as Branin and Beale), as their learned signal variance is usually large and will not trigger the $\delta_3$ condition. For functions that are better modeled by a regular non-skewed GP (such as Hartmann and Ackley), when given the bound information, the learned signal variance will be small, usually in the range from $10^{-4}$ to $10^{-1}$. Therefore, to ensure we can discard all non-informative bound information, we recommend setting $\delta_3$ to be moderately larger. In the Hartmann and Ackley functions, we observe that too small a value (e.g., $\delta_3 = 0.1^2$) can slightly worsen the performance.

### 5.3 Using bound information in model training

When the bound information is available, we use MAP to allow use of this information, and uncertainty level and variance threshold to allow BABO to ignore the bound information if it would lead to a model mismatch. A possible question is whether it is necessary to handle the prior-data conflict explicitly. To answer this question, we compare the following settings with SlogTEI:

- Using MAP only

- Using MAP and the uncertainty level

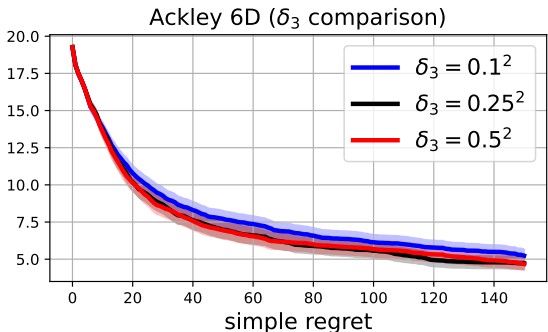

Figure 10: Analysis on the Ackley function demonstrates that setting $\delta_3$ to excessively small values may lead to marginally degraded performance.

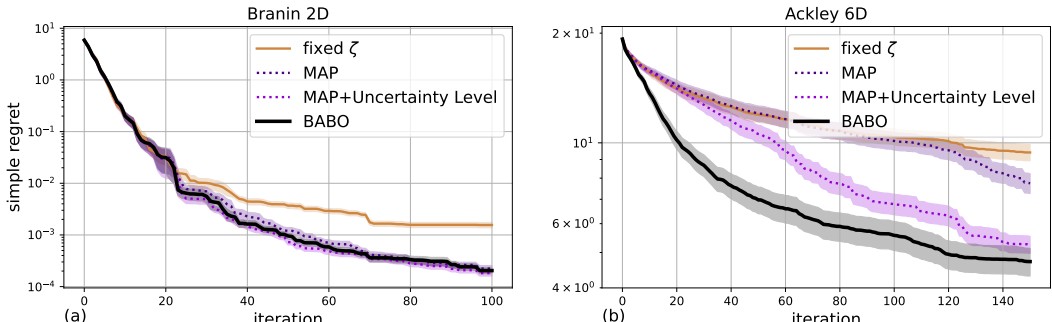

Figure 11: Ablation Study on How to Use Prior Information: For the Branin function, where the known lower bound is quite accurate, all methods (MAP, MAP+Uncertainty Level, BABO) demonstrate comparable performance. In contrast, for the Ackley function, the optimal fitted value of $-\hat{\zeta}$ is substantially negative, creating a significant prior-data conflict with the known lower bound $f^b = 0$. In this case, disregarding the prior bound information in the surrogate model leads to superior performance.

We do experiments on two objective functions: Branin and Ackley. The results are shown in Figure 11. On the Branin function, their performance is almost identical at the beginning while MAP-only is slightly worse later on. This is because when we have enough data, we can estimate the underlying lower bound accurately even if the prior information can be slightly misleading. For the Ackley function, MAP is worse than MAP+uncertainty level which in turn is worse than BABO. This is because the best fitting $-\hat{\zeta}$ of SlogGP is very negative, so $f^b = 0$ conflicts with observation data and not using the bound information is a better choice.

## 5.4 The influence of lower bound information in SlogGP-based BO

We first test how different known lower bound information $f^b$ influences BABO performance. We test on three objective functions: Branin, Beale and Hartmann. Our experimental results in Figure 12 demonstrate that BABO's performance consistently improves as the known lower bound $f^b$ converges toward the true minimum value $f^*$.

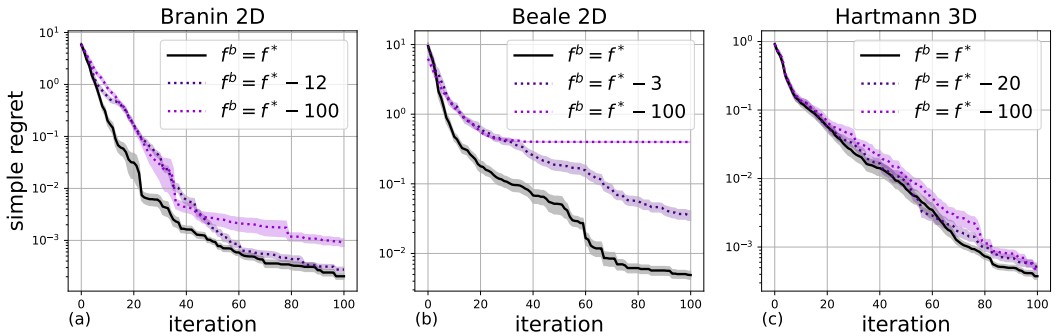

Figure 12: BABO with different $f^b$: BABO's performance improves as the known lower bound $f^b$ approaches the minimum value $f^*$.

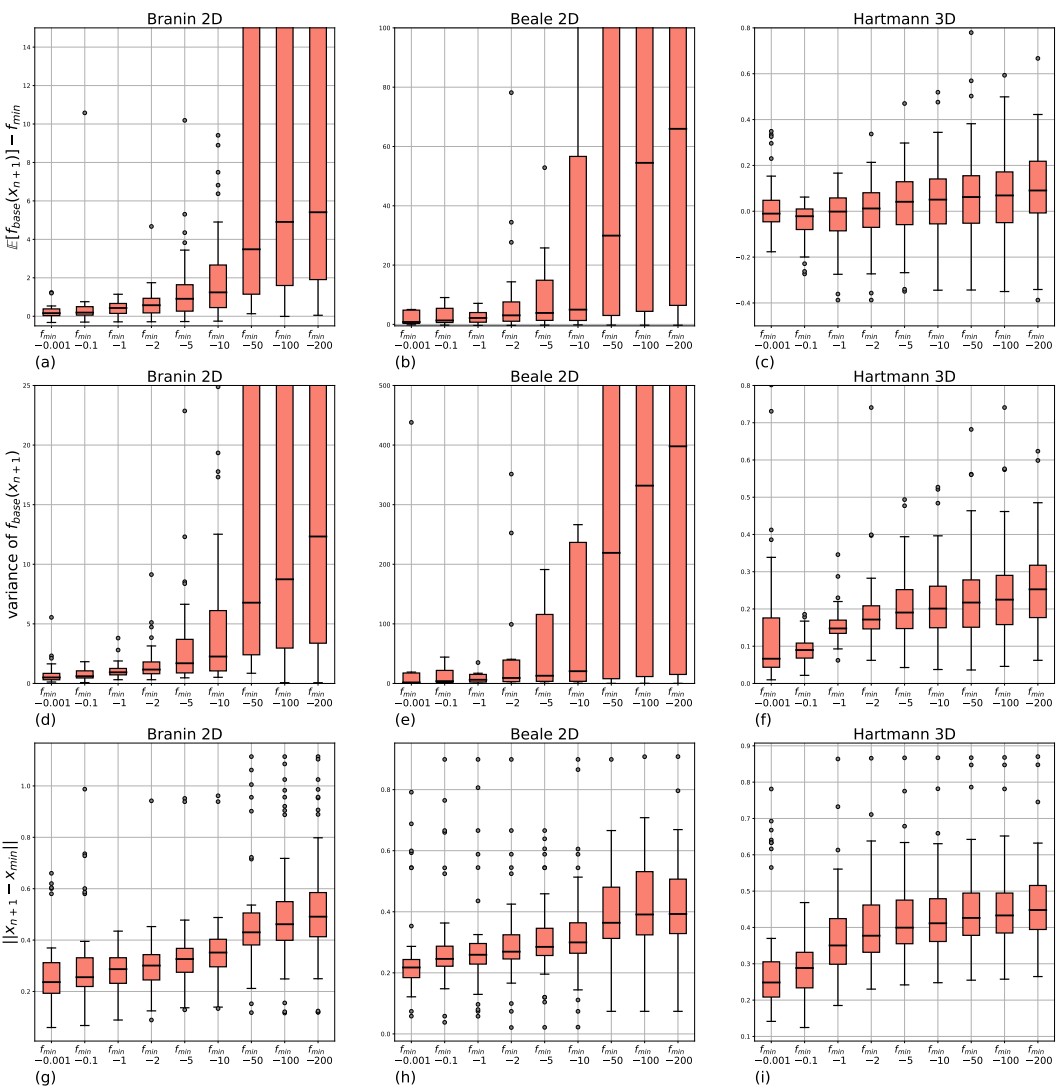

Figure 13: Exploitation and Exploration Analysis: When $-\hat{\zeta}$ is higher (towards the left of each figure), BABO prefers more exploitation; When $-\hat{\zeta}$ is lower, BABO prefers more exploration. Top row shows $\mathbb{E}[f_{base}(\mathbf{x}_{n+1})] - f_{min}$, middle row shows variance at next sample location, and bottom row shows distance of new sample location from current best.

BABO consists of two parts: SlogGP$^b$ and SlogTEI. Although it is clear that the closer $f^b$ and $f^*$, SlogTEI should be better, it is not that obvious how the prior information influences the surrogate model. The influence of a given lower bound in SlogGP on BABO performance is clear when there is no model mismatch: when $f^b$ is close to the underlying $-\zeta$, the prior information of the lower bound can help to learn the model lower bound quickly. On the other hand, when $f^b$ is far away from the best fitting $-\zeta$, this information can be misleading, which is why we reduce the impact of the prior information if a large gap is found.

If there is model mismatch (which is common in practice), the influence of lower bound information on the surrogate model is less clear. However, we found that the estimated parameter $-\hat{\zeta}$ influences the balance between exploitation and exploration. Specifically, a higher value of $-\hat{\zeta}$ corresponds to a greater emphasis on exploitation, as the current best value in the underlying GP model $g_{min} = \ln(f_{min} + \hat{\zeta})$ will be more negative as $-\hat{\zeta}$ increases. Hence, BO is more likely to search in close proximity to $\mathbf{x}_{min}$, focusing more on exploitation. Conversely, if $-\hat{\zeta}$ is lower and far away from $f_{min}$, then BO will do more exploration.

In order to demonstrate this effect, we study the relationship between $-\hat{\zeta}$ and the properties of the next sample chosen. Our base model is SlogGP$^b$ with $-\hat{\zeta} = f_{min} - 1$.

Figures 13(a)-(c) show that the gap between the mean of the next evaluation value and $f_{min}$, i.e. $\mathbb{E}[f_{base}(\mathbf{x}_{n+1})] - f_{min}$ increases as $-\hat{\zeta}$ is lowered. Similarly, Figures 13(d)-(f) show that the predicted variance of $f_{base}(\mathbf{x}_{n+1})$ increases as $-\hat{\zeta}$ is lowered.

Finally, Figures 13(g)-(i) present the relationship between Euclidean distance $||\mathbf{x}_{n+1} - \mathbf{x}_{min}||$ and $-\hat{\zeta}$. A more local search near $\mathbf{x}_{min}$ in case of larger $-\hat{\zeta}$ is an indication of more exploitation in BO.

## 6 Conclusion

We proposed BABO, a Bayesian optimization algorithm that is able to leverage prior information about the optimal value $f^*$ to achieve better solutions and a higher sample-efficiency. BABO uses the tailored surrogate model SlogGP and the acquisition criterion Shifted Logarithmic Truncated Expected Improvement (SlogTEI), which is an extension of Expected Improvement that takes the prior information into account. Experimental results demonstrate that BABO benefits from the prior information and outperforms previous approaches. Moreover, we find that even without prior information about $f^*$, the SlogGP model often performs better than the commonly used GP model when combined with EI. For future work, we will investigate the potential of the SlogGP model further for a wider range of scenarios.

We observe that BABO sometimes chooses a large absolute value for the parameter $\zeta$ of the SlogGP model, which means that the modeled function is 'pushed away' from the known lower bound on the optimal value. This is because the parameter fitting trades off model fit with using the prior information about $f^*$. Although this effect is mitigated by incorporating the information about $f^*$ also in the acquisition criterion SlogTEI, we wonder if there is a more suitable way of jointly modeling the unknown objective and the bound on its optimal value.

### Acknowledgments

We sincerely thank the anonymous reviewers for their valuable comments and constructive suggestions, which have greatly helped improve the quality of this paper. The first author gratefully acknowledges support by the Engineering and Physical Sciences Research Council through the Mathematics of Systems II Centre for Doctoral Training at the University of Warwick (reference EP/S022244/1).

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

## A   Appendix

### A.1   $\alpha^{\mathrm{MES}^b}\left(\mathbf{x}\mid f^b\right)$ shares the same maximizer with $\mathbb{P}(f(\mathbf{x}) < f^b)$

$\mathrm{MES}^b$ shares the same maximizer with the acquisition function $\mathbb{P}(f(\mathbf{x}) < f^b)$, i.e. $\mathrm{argmax}_{\mathbf{x}\in\mathcal{X}}\alpha^{\mathrm{MES}^b}\left(\mathbf{x}\mid f^b\right) = \mathrm{argmax}_{\mathbf{x}\in\mathcal{X}}\mathbb{P}(f(\mathbf{x}) < f^b)$. See examples in Figure 14.

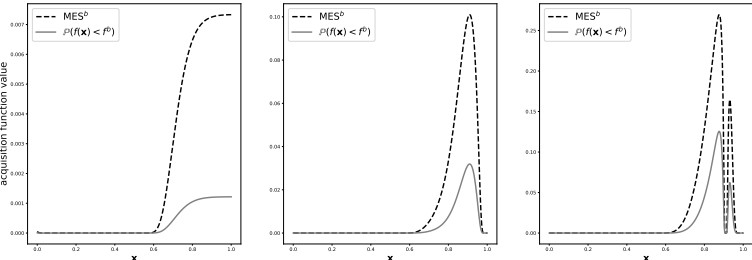

Figure 14: $\mathrm{MES}^b$ and $\mathbb{P}(f(\mathbf{x}) < f^b)$

*Proof.* For simplicity, we use $\gamma$ to represent $\gamma\left(\mathbf{x}, f^b\right) = \frac{\mu(\mathbf{x})-f^b}{\sigma(\mathbf{x})}$ for the following analysis.

Firstly, $\mathbb{P}(f(\mathbf{x}) < f^b) = \Phi(\frac{f^b-\mu(\mathbf{x})}{\sigma(\mathbf{x})}) = 1 - \Phi(\gamma)$, so it is monotonically decreasing with $\gamma$.

The next step is to prove that $\mathrm{MES}^b$ is also decreasing with $\gamma$. We look at $\alpha^{\mathrm{MES}^b}\left(\mathbf{x}\mid f^b\right) = \frac{\gamma\phi(\gamma)}{2\Phi(\gamma)} - \log\left(\Phi\left(\gamma\right)\right)$. For simplicity, we denote it by $M(\gamma)$. The derivative of $M(\gamma)$ is:

$$M'(\gamma) = \frac{1}{2}\cdot\left(\frac{\phi(\gamma)}{\Phi(\gamma)} + \frac{\gamma\cdot\phi'(\gamma)}{\Phi(\gamma)} - \frac{\gamma\cdot\phi(\gamma)^2}{\Phi(\gamma)^2}\right) - \frac{\phi(\gamma)}{\Phi(\gamma)}$$

$$= \frac{1}{2}\cdot\left(-\frac{\phi(\gamma)}{\Phi(\gamma)} + \frac{\gamma\cdot\phi'(\gamma)}{\Phi(\gamma)} - \frac{\gamma\cdot\phi(\gamma)^2}{\Phi(\gamma)^2}\right)$$

$$= \frac{1}{2}\cdot\left(-\frac{\phi(\gamma)}{\Phi(\gamma)} + \frac{\gamma\cdot-\gamma\phi(\gamma)}{\Phi(\gamma)} - \frac{\gamma\cdot\phi(\gamma)^2}{\Phi(\gamma)^2}\right)$$

$$= \frac{1}{2}\cdot\left(-\frac{\phi(\gamma)}{\Phi(\gamma)} - \frac{\gamma^2\cdot\phi(\gamma)}{\Phi(\gamma)} - \frac{\gamma\cdot\phi(\gamma)^2}{\Phi(\gamma)^2}\right)$$

$$= -\frac{\phi(\gamma)}{2}\cdot\left(\frac{1}{\Phi(\gamma)} + \frac{\gamma^2}{\Phi(\gamma)} + \frac{\gamma\cdot\phi(\gamma)}{\Phi(\gamma)^2}\right)$$

$$= -\frac{\phi(\gamma)}{2\Phi(\gamma)^2}\cdot\left(\Phi(\gamma)(1+\gamma^2) + \gamma\cdot\phi(\gamma)\right)$$

To determine the monotonicity of $M(\gamma)$, we need to determine whether $D(\gamma) = \left(\Phi(\gamma)(1+\gamma^2) + \gamma\cdot\phi(\gamma)\right)$ is positive or negative. It is easy to calculate $D'(\gamma) = 2\phi(\gamma) + 2\gamma\Phi(\gamma)$. Given $D''(\gamma) = 2\Phi(\gamma) > 0$, $D'(\gamma)$ is an increasing function in $(-\infty, \infty)$ and the minimal value is achieved when $\gamma \to -\infty$:

$$\lim_{\gamma\to-\infty} D'(\gamma) = \lim_{\gamma\to-\infty} 2(\phi(\gamma) + \gamma\Phi(\gamma))$$

$$= 2\cdot\lim_{\gamma\to-\infty}\gamma\Phi(\gamma)$$

$$= 2\cdot\lim_{\gamma\to-\infty}\frac{\Phi'(\gamma)}{(\frac{1}{\gamma})'}$$

$$= 2\cdot\lim_{\gamma\to-\infty}-\phi(\gamma)\gamma^2$$

$$= 0$$

Hence, $D'(\gamma) > 0$ for $\gamma \in (-\infty, \infty)$, so $D(\gamma)$ is an increasing function. Similarly, $D(\gamma) \to 0$ when $\gamma \to -\infty$, so $D(\gamma) > 0$ for $\gamma \in (-\infty, \infty)$. Hence, the derivative of $M(\gamma)$ is negative, i.e. $M(\gamma)$ is a decreasing function of $\gamma$.

Considering $\gamma$ is a function of $\mathbf{x}$, $\mathbb{P}(f(\mathbf{x}) < f^b)$ and $\alpha^{\mathrm{MES}^b}(\mathbf{x} \mid f^b)$ share the same monotonicity with respect to $\mathbf{x}$, they share the same maximizer. $\qquad\square$

Figure 15 shows the decreasing relationship between $\gamma$ and $\mathrm{MES}^b$. Consider $\mathrm{MES}^b$ and $\mathbb{P}(f(\mathbf{x}) < f^b)$ are both decreasing with $\gamma$, they share the same maximizer $\gamma^*$.

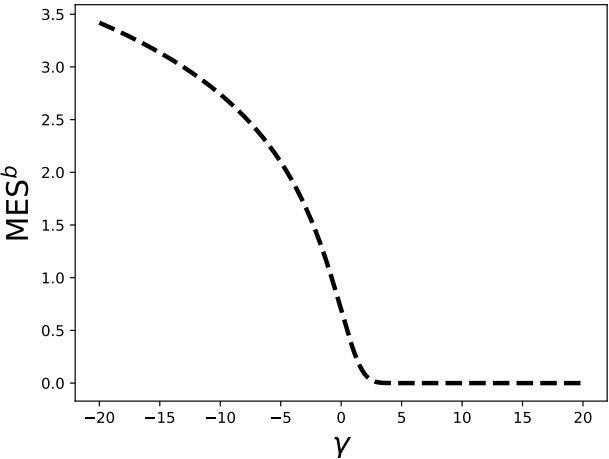

Figure 15: $\mathrm{MES}^b$ is decreasing with $\gamma$. As $\mathrm{MES}^b$ and $\mathbb{P}(f(\mathbf{x}) < f^b)$ are both decreasing with $\gamma$, they share the same maximizer $\gamma^*$.

## A.2 Adapting the acquisition functions EI and PI for the SlogGP model

In the following section, we propose two adaptations of popular acquisition functions to the SlogGP model, Shifted Logarithmic Probability of Improvement (SlogPI) and Shifted Logarithmic Expected Improvement (SlogEI). If the current best value is denoted by $f_{min}$, SlogPI is

$$
\begin{aligned}
\alpha^{\mathrm{SlogPI}}(\mathbf{x}; f_{min}) &= \mathbb{P}(f(\mathbf{x}) \leq f_{min}) \\
&= \Phi\left(\frac{\ln(f_{min} + \zeta) - \mu(\mathbf{x})}{\sigma(\mathbf{x})}\right).
\end{aligned}
$$

The calculation process is shown as follows.

For a solution $\mathbf{x}$ and the current minimal observation $f_{min}$, we denote the predicted mean of $g(\mathbf{x})$ as $\mu(\mathbf{x})$ and variance of $g(\mathbf{x})$ as $\sigma^2(\mathbf{x})$. Due to the normal distribution of $g(\mathbf{x})$, $e^{g(\mathbf{x})}$ (denoted by $Z$) follows a log-normal distribution. Hence, SlogPI can be calculated as follows:

$$
\begin{aligned}
\alpha^{\mathrm{SlogPI}}(x; f_{min}) \quad &= \mathbb{P}(f(\mathbf{x}) \leq f_{min}) \\
&= \mathbb{P}(Z - \zeta \leq f_{min}) \\
&= \mathbb{P}(Z \leq f_{min} + \zeta) \\
&= \Phi(\frac{ln(f_{min} + \zeta) - \mu(\mathbf{x})}{\sigma(\mathbf{x})})
\end{aligned}
$$

The formula for SlogEI and its derivation are presented in Section 3.3. SlogEI is differentiable with $\mathbf{x}$ if $\frac{\partial \mu}{\mathbf{x}_i}$ and $\frac{\partial \sigma}{\mathbf{x}_i}$ exist. The partial derivative is

$$\frac{\partial \alpha^{\mathrm{SlogEI}}}{\partial \mathbf{x}_i} = \frac{\partial \alpha^{\mathrm{SlogEI}}}{\partial \mu} \cdot \frac{\partial \mu}{\partial \mathbf{x}_i} + \frac{\partial \alpha^{\mathrm{SlogEI}}}{\partial \sigma} \cdot \frac{\partial \sigma}{\partial \mathbf{x}_i}.$$

where

$$\begin{aligned}
\frac{\partial \alpha^{\mathrm{SlogEI}}}{\partial \mu} &= -\frac{(f_{min} + \zeta)}{\sigma(\mathbf{x})} \cdot \phi\left(\frac{\ln(f_{min} + \zeta) - \mu(\mathbf{x})}{\sigma(\mathbf{x})}\right) \\
&+ \frac{e^{\mu(\mathbf{x}) + \frac{\sigma^2(\mathbf{x})}{2}}}{\sigma(\mathbf{x})} \cdot \phi\left(\frac{\ln(f_{min} + \zeta) - \mu(\mathbf{x}) - \sigma^2(\mathbf{x})}{\sigma(\mathbf{x})}\right) \\
&- e^{\mu(\mathbf{x}) + \frac{\sigma^2(\mathbf{x})}{2}} \cdot \Phi\left(\frac{\ln(f_{min} + \zeta) - \mu(\mathbf{x}) - \sigma^2(\mathbf{x})}{\sigma(\mathbf{x})}\right)
\end{aligned}$$

and

$$\begin{aligned}
\frac{\partial \alpha^{\mathrm{SlogEI}}}{\partial \sigma} &= -\frac{(f_{min} + \zeta)}{\sigma^2(\mathbf{x})} \cdot \phi\left(\frac{\ln(f_{min} + \zeta) - \mu(\mathbf{x})}{\sigma(\mathbf{x})}\right) \cdot (\ln(f_{min} + \zeta) - \mu(\mathbf{x})) \\
&+ e^{\mu(\mathbf{x}) + \frac{\sigma^2(\mathbf{x})}{2}} \cdot \phi\left(\frac{\ln(f_{min} + \zeta) - \mu(\mathbf{x}) - \sigma^2(\mathbf{x})}{\sigma(\mathbf{x})}\right) \cdot (\frac{\ln(f_{min} + \zeta) - \mu(\mathbf{x})}{\sigma^2(\mathbf{x})} + 1) \\
&- \sigma(\mathbf{x}) e^{\mu(\mathbf{x}) + \frac{\sigma^2(\mathbf{x})}{2}} \cdot \Phi\left(\frac{\ln(f_{min} + \zeta) - \mu(\mathbf{x}) - \sigma^2(\mathbf{x})}{\sigma(\mathbf{x})}\right).
\end{aligned}$$

### A.3 Acquisition Function Visualization

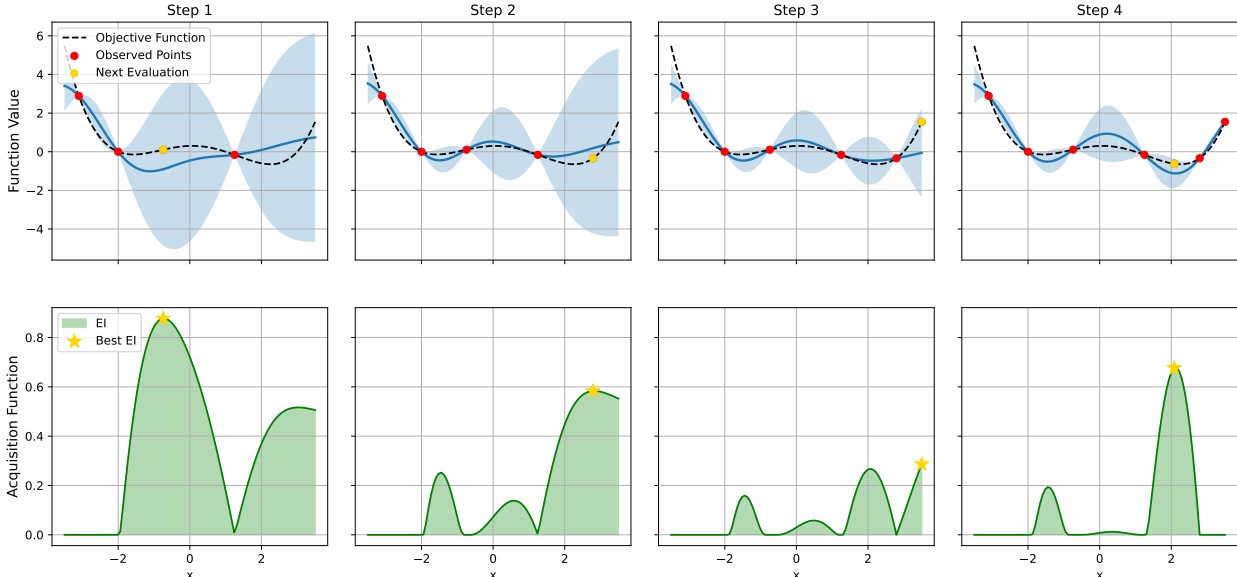

Figure 16: Example of data acquisition of first 4 iterations of BO using GP+EI. Upper figures shows surrogate model and true objective function, lower figures show acquisition function. Three initial points.

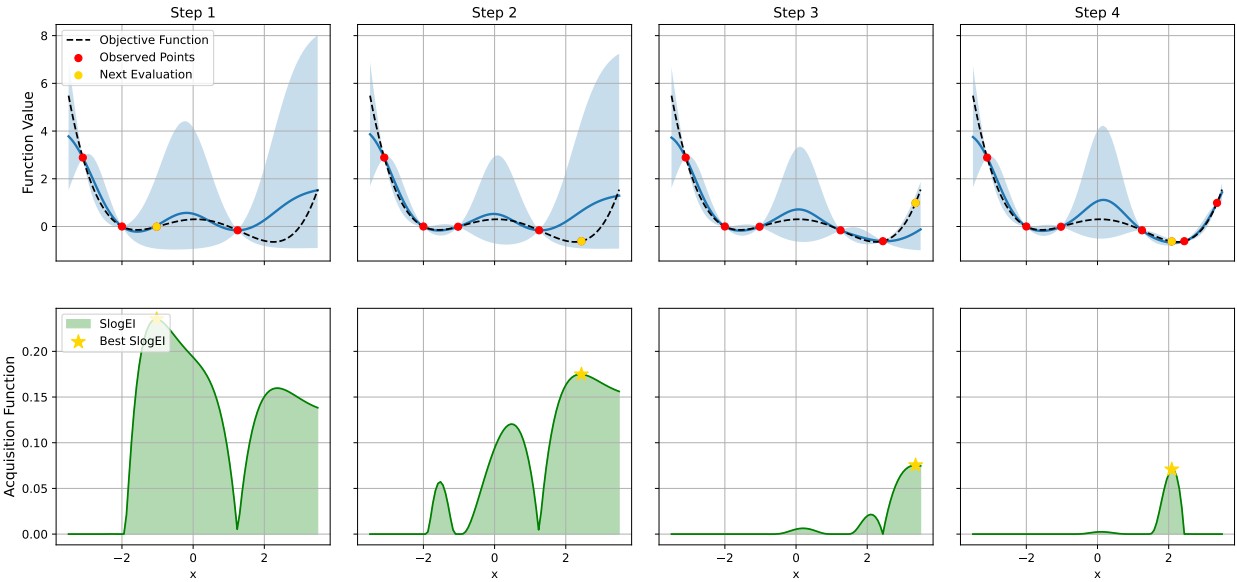

Figure 17: Example of data acquisition of first 4 iterations of BO using SlogGP+SlogEI. Upper figures shows surrogate model and true objective function, lower figures show acquisition function. Three initial points.

## A.4 Proof of Theorem 3.1

For convenience, we repeat Theorem 3.1 first before stating the proof.

**Theorem 3.1.** Define a SlogGP $f(\mathbf{x}) = e^{g(\mathbf{x})+\mu_g} - \zeta$ with $g(\mathbf{x})$ a Gaussian process with zero mean, zero noise and a covariance function $K$ with signal variance $\sigma_g^2$ and other hyperparameters $\theta_g$.

Then for any GP $h(\mathbf{x})$ with mean $\tilde{\mu}_g$, signal variance $\tilde{\sigma}_g$, and other hyperparameters $\tilde{\theta}_g$, we can have the SlogGP $f(\mathbf{x})$ converge to $h(\mathbf{x})$ in distribution for any $\mathbf{x} \in \mathcal{X}$ by setting

$$\begin{cases} \mu_g = \ln\left(\zeta + \tilde{\mu}_g\right) \\ \theta_g = \tilde{\theta}_g \\ \sigma_g = \dfrac{\tilde{\sigma}_g}{\zeta + \tilde{\mu}_g} \end{cases}$$

and letting $\zeta \to \infty$, i.e.,

$$\lim_{\zeta \to \infty} \mathcal{P}_{f(\mathbf{x})}(z) = \mathcal{P}_{h(\mathbf{x})}(z) \quad \forall z \in (-\zeta, \infty), \ \forall \mathbf{x} \in \mathcal{X},$$

where $\mathcal{P}(\cdot)$ denotes the probability density function of a random variable.

*Proof.* Given a SlogGP $f(\mathbf{x}) = e^{g(\mathbf{x})+\mu_g} - \zeta$ with $g(\mathbf{x})$ a Gaussian process with zero mean, zero noise and a covariance function $K$ with signal variance $\sigma_g^2$ and other hyperparameters $\theta_g$. We set $\sigma_g, \theta_g, \mu_g$ such that the following equations:

$$\begin{cases} \mu_g = \ln\left(\zeta + \tilde{\mu}_g\right) \\ \theta_g = \tilde{\theta}_g \\ \sigma_g = \dfrac{\tilde{\sigma}_g}{\zeta + \tilde{\mu}_g} \end{cases}$$

are satisfied for some constants $(\tilde{\sigma}_g, \tilde{\mu}_g, \tilde{\theta}_g) \in \Theta$.

According to the definition $e^x = \sum_{k=0}^{\infty} \frac{x^k}{k!}$, we have

$$
\begin{aligned}
f(\mathbf{x}) \quad &= e^{\mu_g} e^{g(\mathbf{x})} - \zeta \\
&= e^{\mu_g} \cdot (1 + g(\mathbf{x}) + \frac{1}{2} g^2(\mathbf{x}) + \frac{1}{6} g^3(\mathbf{x})) + ...) - \zeta \\
&= e^{\mu_g} g(\mathbf{x}) + e^{\mu_g} - \zeta + e^{\mu_g} (\frac{1}{2} g^2(\mathbf{x}) + \frac{1}{6} g^3(\mathbf{x})) + ...).
\end{aligned}
$$

In the formula, we distinguish three different parts: $e^{\mu_g} g(\mathbf{x})$, $e^{\mu_g} - \zeta$ and $e^{\mu_g}(\frac{1}{2} g^2(\mathbf{x}) + \frac{1}{6} g^3(\mathbf{x}) + ...)$.

In terms of the first part, $g(\mathbf{x})$ is a Gaussian process with zero noise, zero mean and kernel with signal variance $\sigma_g^2$ and other hyperparameters $\theta_g$. Hence, $e^{\mu_g} g(\mathbf{x})$ is a Gaussian process with zero noise, zero mean and kernel with signal variance $e^{2\mu_g} \sigma_g^2$ and other hyperparameters $\theta_g$.

The second part $e^{\mu_g} - \zeta$ is the constant $\tilde{\mu}_g$ as we set $\mu_g = \ln(\zeta + \tilde{\mu}_g)$.

For the third part, we first look at $e^{\mu_g} g^2(\mathbf{x})$. We have $e^{\mu_g} g^2(\mathbf{x}) = (e^{\frac{\mu_g}{2}} g(\mathbf{x}))^2$, so $e^{\mu_g} g^2(\mathbf{x})$ is the square of a Gaussian process with signal variance $e^{\mu_g} \sigma_g^2$ and mean 0. When $\zeta \to \infty$, we have $e^{\mu_g} \sigma_g^2 \to 0$, so $\forall \mathbf{x}$

$$
\lim_{\zeta \to \infty} \mathbb{P}\left( \left| e^{\mu_g} g^2(\mathbf{x}) - 0 \right| > \varepsilon \right) = 0.
$$

For other terms $e^{\mu_g} g^k(\mathbf{x})$ in the third part, we can view them as $(e^{\frac{\mu_g}{k}} g(\mathbf{x}))^k$, so similarly, we have $\lim_{\zeta \to \infty} \mathbb{P}\left( \left| e^{\mu_g} g^k(\mathbf{x}) - 0 \right| > \varepsilon \right) = 0$ for $k \geq 3, \forall \mathbf{x}$ and $\forall \varepsilon$.

Therefore, for $\forall \mathbf{x}$ and $\forall \varepsilon$,

$$
\lim_{\zeta \to \infty} \mathbb{P}\left( \left| f(\mathbf{x}) - \tilde{g}(\mathbf{x}) \right| > \varepsilon \right) = 0,
$$

where $\tilde{g}(\mathbf{x}) = e^{\mu_g} g(\mathbf{x}) + e^{\mu_g} - \zeta$ is a Gaussian process with mean $\tilde{\mu}_g$, zero noise and covariance function $K$ with signal variance $\tilde{\sigma}_g^2$ and other hyperparameters $\tilde{\theta}_g$.

Therefore, we have

$$
\lim_{\zeta \to \infty} \mathcal{P}_{f(\mathbf{x})}(z) = \mathcal{P}_{\tilde{g}(\mathbf{x})}(z) \quad \forall z \in (-\zeta, \infty), \ \forall \mathbf{x} \in \mathcal{X},
$$

where $\mathcal{P}(\cdot)$ denotes the probability density function of a random variable.

Given that $\tilde{g}$ and $h$ share the same parameters, they share the same posterior distribution, i.e.

$$
\mathcal{P}_{\tilde{g}(\mathbf{x})}(z) = \mathcal{P}_{h(\mathbf{x})}(z) \quad \forall z \in (-\zeta, \infty), \ \forall \mathbf{x} \in \mathcal{X},
$$

Hence, we have

$$
\lim_{\zeta \to \infty} \mathcal{P}_{f(\mathbf{x})}(z) = \mathcal{P}_{h(\mathbf{x})}(z) \quad \forall z \in (-\zeta, \infty), \ \forall \mathbf{x} \in \mathcal{X},
$$

$\square$

### A.5 Experimental Settings (Within-model Test, Synthetic Function Test and Real-world Benchmark Test)

The experimental setting is as follows. For each $d$-dimensional test function, we sample $4d$ initial points from a Latin hypercube design. The input domain is scaled to $[0, 1]^d$. For GP-based methods, function values are standardized (scaled and centralized) and for SlogGP-based methods, function values are scaled and the centralizing is done in model training. As kernel we use the squared exponential kernel: $K(\mathbf{x}_a, \mathbf{x}_b) = \sigma^2 \exp\left( -\frac{\|\mathbf{x}_a - \mathbf{x}_b\|^2}{2\ell^2} \right)$. For acquisition function optimization, we use restart L-BFGS-B in scipy. The restart time and initial samples (restart number $3d$ and initial sample $30d$) and L-BFGS-B options are the same for all acquisition functions. We compare the performance of BABO with other benchmark algorithms across various test functions. The benchmark algorithms we consider are Random, EI, TEI, MES[b], OBCGP and ERM.

We consider a noiseless setting in our theoretical analysis. In practice, a small positive noise variance is typically introduced to ensure numerical stability. Since the estimated signal variance $\hat{\sigma}_g^2$ in SlogGP can vary substantially, ranging from near-zero to several hundred, we adopt an adaptive noise variance proportional to the estimated signal variance: $\sigma_{\text{noise}}^2 = 10^{-5} \cdot \hat{\sigma}_{g,n-1}^2$ The initial noise variance is set to $6 \cdot 10^{-6}$. We maintain these noise parameter settings across all comparative methods to ensure fair comparison.

Details of test functions are shown in the table below.

Table 3: Test Function Information

| Test Function | Optimal Value | Search Space |
|---|---|---|
| GP-generated functions (2D) | - | $[0., 1.]^2$ |
| SlogGP-generated functions (2D) | - | $[0., 1.]^2$ |
| Beale (2D) | 0 | $[-4.5, 4.5]^2$ |
| Branin (2D) | 0.397887 | $[[-5., 10.], [0., 15.]]$ |
| SixHumpCaml (2D) | -1.0316 | $[[-3., 3.], [-2., 2.]]$ |
| Levy (2D) | 0 | $[[-10., 10.], [-10., 10.]]$ |
| Hartmann (3D) | -3.86278 | $[0., 1.]^d$ |
| DixonPrice (4D) | 0 | $[-10., 10.]^d$ |
| Rosenbrock (4D) | 0 | $[-2.048, 2.048]^d$ |
| Ackley (6D) | 0 | $[-32.768, 32.768]^d$ |
| Powell (8D) | 0 | $[-4., 5.]^d$ |
| StyblinskiTang (10D) | -391.6599 | $[-5., 5.]^d$ |
| PDE Variance (4D) | - | $[[0.1., 5.], [0.1, 5.], [0.01, 5.], [0.01, 5.]]$ |
| Robot Push (4D) | 0 | $[[-5., 5.], [-5., 5.], [0., 2\pi], [0., 300.]]$ |
| XGBoost Hyperparameter Tuning (6D) | - | $[[0., 10.], [0., 10.], [5., 15.], [1., 20.], [0.5, 1.], [0.1, 1.]]$ |

In addition, in terms of within-model in Section 3.4, for GP-generated functions, signal variance $\sigma_g^2 = 2$, lengthscale $l = 0.1$ and mean 0. For SlogGP-generated functions, signal variance $\sigma_g^2 = 1.2$, lengthscale $l = 0.1$, shift $\zeta = 30$ and mean 0.5.

We also had a cross-validation test for GP and SlogGP. The repetition number is 50. In each experiment, 40 initial training points are sampled randomly. We test the gap between model prediction mean at a random $\mathbf{x}$ and its value $f(\mathbf{x})$.

### A.6   More experiments: batched BABO

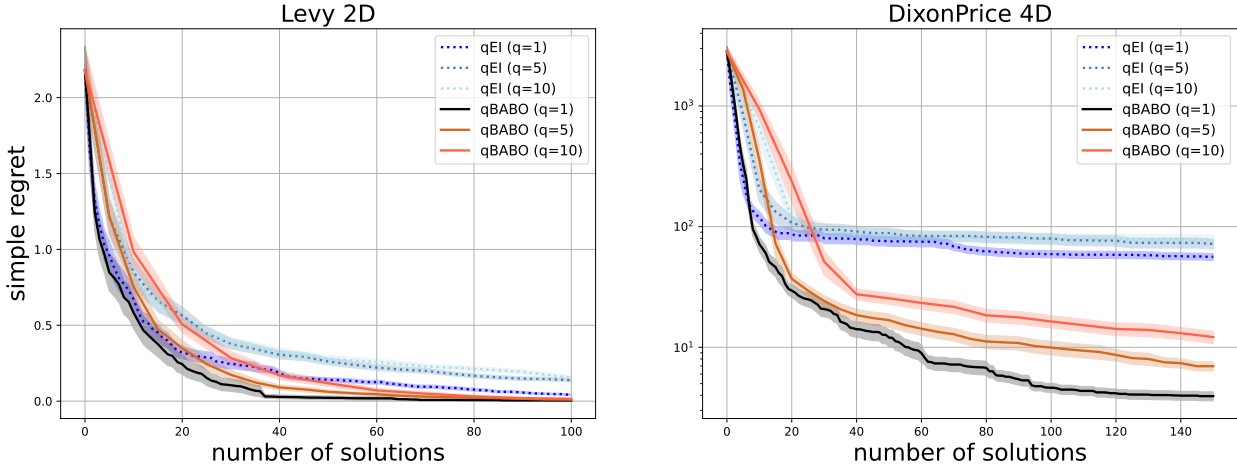

Figure 18: Batched BO: We evaluated the performance across varying batch sizes (q = 1, 5, and 10) for both test problems. The experimental results demonstrate that qBABO consistently outperforms qEI across all batch size configurations.

