# OpenReview forum: "Respecting the limit: Bayesian optimization with a bound on the optimal value"
_TMLR — Accepted by TMLR_

### Review · Reviewer_bJPZ · 2025-01-08

**Summary Of Contributions:**

The paper proposes an extension of Bayesian optimization that combines log transformations of the target function with incorporating a bound on the optimum value in the model. The bound on the optimum value is a realistic assumption, because in machine learning we often know the best possible value, for example, accuracy is bounded between 0 and 1. The paper demonstrates superior results compared to standard Bayesian optimization on 8 synthetic functions, one XGBoost hyperparameter optimization problem and the 4D robot push problem. Furthermore, the paper contains a thorough ablation study that shows the benefits of the individual contributions.

**Audience:**

Yes

**Claims And Evidence:**

Yes

**Requested Changes:**

* Please provide code for your method and the experiments.
* Please make the paper more self-contained:
    * Provide the mean and variance prediction equations of the SlogGP.
    * Why is the acquisition function derivation in the Appendix? This is a journal paper, I would assume such central information in the main paper. Additionally, it would be great if the new acquisition functions could be contrasted with general EI and PI as described by [Brochu et al. (arXiv, 2010)](https://arxiv.org/abs/1012.2599).
* Please add more benchmark tasks. In the past years, researchers have published and used various new benchmarks, see for example [HPOBench](https://datasets-benchmarks-proceedings.neurips.cc/paper/2021/hash/93db85ed909c13838ff95ccfa94cebd9-Abstract-round2.html) or the recent comparison of [BNN](https://openreview.net/forum?id=SA19ijj44B).
* The paper should discuss the related work [unexpected improvement to expected improvement](https://proceedings.neurips.cc/paper/2023/hash/419f72cbd568ad62183f8132a3605a2a-Abstract-Conference.html)
* The method introduces a total of 3 additional hyperparameters. However, there is no further analysis of this, and the paper should contain a sensitivity analysis on them.
* The Ablation misses SlogGP+EI.

## Minor / Questions

* Page 2: instead of "For Max-value Entropy Search (MES) introduced by Wang & Jegelka (2017)" I suggest writing "For Max-value Entropy Search (MES; Wang & Jegelka, 2017)", which introduces the reading flow because one does not have to focus on the author names, which add little additional information.
* How does the statement "In addition, we show in Appendix A.1 that MESb shares the same maximizer with the acquisition function P(f (x) < f b), i.e. argmaxx∈X αMESb (x | f b) = argmaxx∈X P(f (x) < f b)." relate to SlogEI?
* The text describing Figure 1 states "The hyperparameters of both methods are fit to data.", and I am wondering to which data the hyperparameters are fit? If we have a known GP, why don't we fit the BO surrogate to the GP hyperparameters?
* Regarding the experimental setup (A4) I have the following questions: is the fixed variance a standard procedure in BO? So far, I have not seen this, and especially in the case of noiseless functions I wonder about the effect of adding a fixed noise. Why not estimate the noise too, or set it to zero (or near-zero) in the case of synthetic functions?
* Figure 11 is horribly typeset, and the caption is below the page number.
* The ticklabels are hard to read and their font size should be increased.
* Section 3.5 contains pseudo-code and various additions to the acquisition function. I suggest moving the pseudo-code to its own subsection, as I find it a bit confusing and did not expect it here. Also, I would appreciate line numbers for algorithms.
* The equation in 3.1 that gives the negative log likelihood should be abbreviated by NLL and not L, and the equation should end with a comma.

**Strengths And Weaknesses:**

## Strengths

* Paper builds on reasonable assumptions in practical applications.
* Strong overall performance.
* Good ablation studies.

## Weaknesses

* The submission does not come with code, and code is only promised upon acceptance.
* Rather limited set of experiments.

---

> ### Author Response · Authors · 2025-01-17
> **Response to Reviewer bJPZ Comments**
>
> We appreciate the constructive feedback from the reviewer. In the following, we respond to the various requested changes and minor questions. We will modify the paper accordingly, although this will take a bit more time.
>
> __Critical Adjustments:__
>
> __Please provide code for your method and the experiments.__
>
> The code should be available in the supplementary material. In the paper we mentioned that it would be released after acceptance, which we put in because we also published the paper on arXiv before submitting it to TMLR. The code can also be downloaded from  https://anonymous.4open.science/r/TMLR-BABO-code-06F0/
>
> __Please make the paper more self-contained:__
>
> As you suggested, we will move the derivation of the acquisition function into the main paper. We would also be happy to provide a visualisation of the GP model and acquisition function over a few steps.
>
> __Please add more benchmark tasks. In the past years, researchers have published and used various new benchmarks, see for example HPOBench or the recent comparison of BNN.__
>
> Ideally we would use some benchmark problems that have a natural known bound. We have run some additional experiments on the XGBoost problem for Wine Quality, Banknotes and Breast Cancer detection (see https://anonymous.4open.science/r/TMLR-BABO-code-06F0/XGBoost_task.pdf), and we are happy to include another one.
>
> __The paper should discuss the related work unexpected improvement to expected improvement__
>
> LogEI is a way to deal with numerical instabilities when optimizing the acquisition function. In our current comparison, all methods (except OBCGP) use the same codebase and the same way of optimising the acquisition function. We expect that BABO as well as EI would benefit from enhanced numerical stability coming from the logEI idea and tricks.
>
> __The method introduces a total of 3 additional hyperparameters. However, there is no further analysis of this, and the paper should contain a sensitivity analysis on them.__
>
> Based on our experimental  observations, the setting of these hyperparameters has little impact, as long as they are roughly in the area suggested. We are currently running a more thorough experimental sensitivity analysis and will upload it when it is completed or include it in the final paper.
>
> __The Ablation misses SlogGP+EI.__
>
> SlogEI is the canonical adaptation of EI for the SlogGP surrogate model. SlogGP alters the distribution of the uncertainty around a GP prediction, it is no longer normal as is assumed by EI, and thus combining it with standard EI would be a mismatch.
>
> __Minor / Questions__
>
> __How does the statement "In addition, we show in Appendix A.1 that MESb shares the same maximizer with the acquisition function $P(f (x) < f^b)$" relate to SlogEI?__
>
> This part is not directly related to SlogGP or SlogEI. The connection arose during our evaluation, as we initially planned to use $P(f(x)<f^b)$ as a benchmark algorithm. However, we discovered that it behaves identically to MES$^b$. Given this finding, we believe it may be of interest to some readers.
>
> __The text describing Figure 1 states "The hyperparameters of both methods are fit to data.", and I am wondering to which data the hyperparameters are fit? If we have a known GP, why don't we fit the BO surrogate to the GP hyperparameters?__
>
> We use a GP-generated test problems, but don’t make use of the information on the hyperparameters, but instead learn them. This is because we couldn’t directly use the SlogGP model parameters for a GP model and vice versa, and we want to demonstrate how the model can fit the data. The data we use for fitting is the data collected during the BO run.
>
> __Regarding the experimental setup (A4) I have the following questions: is the fixed variance a standard procedure in BO? So far, I have not seen this, and especially in the case of noiseless functions I wonder about the effect of adding a fixed noise. Why not estimate the noise too, or set it to zero (or near-zero) in the case of synthetic functions?__
>
> In noiseless BO, the noise variance can be set to zero. However, as you mentioned, to ensure numerical stability, a common practice is to fix a small noise variance. A consideration here is that the signal variance of $g$ in SlogGP can be very small or relatively large, such as $10^{-3}$ and $4$. In such cases, setting the noise variance to a fixed value like $10^{-5}$ can make the SlogGP’s relative noise level different for different signal variance. To address this, we set the noise variance as a fixed fraction of the signal variance in both SlogGP and GP. However, allowing SlogGP and GP to learn the noise variance does not significantly impact the results.
>
> We will fix other minor comments.

---

### Review · Reviewer_oC82 · 2025-01-11

**Summary Of Contributions:**

The paper introduces Bound-Aware Bayesian Optimization (BABO), a novel approach to Bayesian Optimization (BO) that leverages prior information on the achievable bounds of the objective function. Specifically:
- The authors propose a new surrogate model, SlogGP, which incorporates a shift parameter (ζ) to encode bound information, thus improving model expressiveness.
- A tailored acquisition function, SlogTEI, is developed to exploit bound information effectively, improving upon standard Expected Improvement (EI).
- The paper discusses mechanisms for handling mismatches between bound priors and observed data, ensuring robust performance.
- Empirical studies on synthetic and real-world benchmarks validate the proposed method, demonstrating significant performance gains over existing techniques.

**Audience:**

Yes

**Claims And Evidence:**

Yes

**Requested Changes:**

# Critical Adjustments
As mentioned in Weaknesses, the following modifications will enhance the contributions of this paper.
- Analyze hyperparameter sensitivity including experiments to evaluate how variations in $\delta_1, \delta_2, \delta_3$ impact BABO's performance, and discuss practical guidelines for selecting these hyperparameters in diverse scenarios.
- Some discussions and experiments on why BABO outperforms standard BO when the prior bound exactly matches the real bound
- Comparison with parabolic and log-transformed GPs, as mentioned in Weaknesses.
- Provide computational complexity analysis and results.

# Minor Suggestions
- Modifying some typos. e.g., Sec. 4 the citation of OBCGP is wrongly inserted into the word "codebase".
- Provide additional real-world examples where bound information is naturally available and impactful.
- Discuss limitations in scenarios where the proposed might fail.

**Strengths And Weaknesses:**

# Strengths
- Novel Contribution: The introduction of SlogGP and SlogTEI is original and addresses a practical need in BO applications where bound information is available.
- Theoretical Analysis and Practical Methodology: The flexibility of SlogGP is proven. The proposed mechanisms (e.g., MAP estimation for ζ, variance thresholds) are practical for handling prior-data conflicts.
- Comprehensive empirical experiments including ablation studies.
- Clarity: The paper is well-structured, with clear explanations of the methodology and results.

# Weaknesses
- Limited Hyperparameter Analysis: The selection and sensitivity of hyperparameters (e.g., $\delta_1, \delta_2, \delta_3$) are not extensively analyzed, which may affect reproducibility.
- Lack of discussions of why BABO outperforms standard BO. My intuition is that a known optimal bound can help achieve a better balance between exploitation and exploration. For instance, it encourages exploration when the current solution is far from the optimal bound, while promoting exploitation when the current solution is close to the optimal bound.
- Comparison with Broader Models: As mentioned in the paper, the proposed SlogGP model can be viewed as combining aspects of a parabolic $f(x) = \frac{1}{2} g^2(x) - \zeta$ and a log-transformed GP $f(x) = e^{g(x)}$. Thus, a comparison with these models using the proposed Truncated EI (TEI) could further illuminate the contribution and potential advantages of the SlogGP model.
- Computational Complexity: The additional computational overhead of SlogGP and SlogTEI compared to standard GPs is not quantified.

---

> ### Author Response · Authors · 2025-01-17
> **Response to Reviewer oC82 Comments**
>
> We appreciate the constructive feedback from the reviewer. In the following, we respond to the various critical adjustments and minor suggestions. We will modify the paper accordingly, although this will take a bit more time.
>
> __Critical Adjustments:__
>
> __Analyze hyperparameter sensitivity including experiments to evaluate how variations in $\delta_1$, $\delta_2$, $\delta_3$ impact BABO's performance, and discuss practical guidelines for selecting these hyperparameters in diverse scenarios.__
>
> Based on our experimental  observations the setting of these hyperparameters has little impact, as long as they are roughly in the area suggested. We are currently running a more thorough experimental sensitivity analysis and will upload it when it is completed or include it in the final paper.
>
> __Some discussions and experiments on why BABO outperforms standard BO when the prior bound exactly matches the real bound.__
>
> When the prior bound exactly matches the real bound this should be the most informative best case scenario. However, even if the prior bound is not a good match, SlogGP can work better than GP as the former can capture skewness.
>
> __Comparison with parabolic and log-transformed GPs, as mentioned in Weaknesses.__
>
> What we meant is that BABO combining aspects of the parabolic and the log-transformed GP was that BABO has the flexibility of the parabolic model (learning the bound), and the guarantee of adhering to the bound as the log-transformed GP. For a log-transformed GP with $\zeta$ fixed to the prior bound, truncation has no additional benefit, as the model guarantees to obey the bound.  The BABO with fixed $\zeta$  in Figure 4 shows the performance of the log-transformed GP (with SlogEI or SlogTEI, it would behave the same). Regarding the combination of the parabolic model with TEI, in their original paper [Nguyen and Osborne, 2020], the authors state that using EI or UCB doesn’t really work in combination with their model: “As the effect of transformation, the predictive uncertainty $\sigma(x)$ of the transformed GP becomes larger than in the case of vanilla GP … This property may let other acquisition functions (e.g., UCB, EI) explore more aggressively than they should.”  If the reviewer feels that nevertheless this should be explored, we would be happy to do so.
>
> __Provide computational complexity analysis and results.__
>
> The computational complexity of BABO matches that of standard BO and thus is dominated by the cost of Bayesian inference in each step of the BO algorithm. The main difference is that BABO has one additional parameter to tune. The difference in runtime is not noticeable. BO is aimed at problems where the function evaluations are computationally expensive, so any small difference in model fitting is negligible.
>
> __Minor suggestions:__
>
> __Provide additional real-world examples where bound information is naturally available and impactful.__
>
> There are indeed many real-world examples where a natural bound is known. For any heat engine, the efficiency cannot exceed the Carnot efficiency, which depends on the temperatures of the heat source and sink. The upper bound of data rate in a communication channel is given by the Shannon-Hartley theorem, which defines the maximum capacity based on bandwidth and signal-to-noise ratio. Latency cannot be reduced below the physical propagation time of a signal. In aerodynamics the drag cannot be reduced below zero because drag represents resistance.
>
> __Discuss limitations in scenarios where the proposed might fail.__
>
> We already show results on the Ackley function where BABO is not the best method, see Figure 4. This may be the case when the underlying function has characteristics quite alien to the SlogGP model such as a very steep funnel to the optimum. We can explain this in the paper.

---

### Review · Reviewer_RefA · 2025-01-15

**Summary Of Contributions:**

The authors propose a novel Bayesian optimization (BO) framework BABO specifically tailored to scenarios when a lower bound on or the minimum value of the objective function is known; though BABO can also be used when such information is not available. BABO utilizes a modification of a Gaussian process, SlogGP, as a surrogate model, which combines a log-transformed GP with a learnable shift parameter. To use SlogGP, whose predictive distribution is no longer normally distributed, the authors adapt the Expected Improvement acquisition function and additionally incorporate the lower bound information, resulting in a novel acquisition function SlogTEI. The authors show that SlogGP with SlogTEI (which together constitute the BABO framework) outperforms GP-based BO and other existing methods on a range of experiments. The authors also perform a range of ablations evaluating the contributions of leveraging bound information. The authors further prove that SlogGP is a more expressive model than the GP.

**Audience:**

Yes

**Claims And Evidence:**

Yes

**Requested Changes:**

### Requested Changes

- Please briefly state if there are any commonly used conditions under which Theorem 3.1 does not hold. For example, can a SlogGP work in a noisy setting?
- 3.4: Please show that the parameters of the $ζ_{prior}$ result in the median / mean of $ζ$ as specified. Why is it desirable to have the median equal to $f_b$ and mean be $\delta_1$ away?
- 3.4: The criterion for $\hat{ζ}$ being far away from the lower bound appears to specify two regions of the CDF of $ζ_{prior}$, a thin slice below $δ_2$ and above $1-δ_2$. Can you please elaborate how this compares the estimated $\hat{ζ}$ to $f_b$?
- 3.4: Please give more intuition for why U is multiplied by the provided expression and define $\phi^{-1}$ after estimating MLE of $ζ$.
- 3.4: Why does signal variance being too small indicate the data-prior conflict? Is it because that would happen when $ζ \rightarrow \inf$, which would indicate it being far away from $f_b$?
- 3.4: How were $\delta_1, \delta_2, \delta_3$ chosen?
- 3.5: Please show why SlogTEI becomes SlogEI for  $-ζ > f_b$.
- Fig. 7: Which method estimated the $\hat{ζ}$ shown with the "no bound information" curve? Plotting the true lower bound on each panel would also be instructive.
- Fig 7: "No bound information" curve often converges to a lower bound than the other curve (which I am assuming to be BABO). Does that mean that the "no bound information" method found a better optimum? In most scenarios, both no-bound and bound methods converge to the same value; what is the interpretation of that? Does this mean that using bound information doesn't confer any utility?
- Please provide a discussion on the difference between $ζ$ being far away from $f_b$ and model mismatch and how this connects to (i) BABO with fixed $ζ$ performing worse than having to learn $ζ$, (ii) BABO learning estimated $ζ$ that is lower than $f_b$ (e.g. Ackley in Fig. 7) and (iii) absorbing both skewness and lower bound information into a single parameter.
- There is inconsistent use of $\hat{ζ}$ and $\tilde{ζ}$ in the manuscript.
- Fig. 10: "Fixed C" in the legend is not referenced anywhere else.
> However, for functions like Beale where the estimated bound converges near the true minimum, this additional bound information proves more beneficial."

What indicates this, the superior performance of SlogGP over GP? But the previous statement suggests that this wouldn't hold on Ackley, yet BABO still performs best on that function.

### Remaining questions:
- Table 1: Why is SlogGP error on the SlogGP test function higher than that of SlogGP on the GP test function? Is a GP function easier to fit that a SlogGP function?
- Fig. 8a: Both GP and SlogGP using TEI perform worse than all other methods. Do you have an intuition about this performance of TEI?
- Can other acquisition functions be modified to be used in SlogGP?
- It could be instructive to also show SlogGP without lower bound information in Fig. 3
- How often is the lower bound information available in practice? Does incorporating more "trivial" bounds such as "distance > 0" still advantageous based on your observations?

### Minor:
- Fig. 1: "with with" typo
- Fig. 3: "l" reads as "1". Might be clearer to spell out left/right
- p. 9: typo in "codebase"
- p. 12: "In the Beale function ... the convergence value of 3" should say "value of -3" according to Fig. 7
- Eq. 3.4 in Algorithm 1 links to Section 3.4 and the equations are not numbered
- Fig. 4: specify GP+EI and GP+TEI in the legend

**Strengths And Weaknesses:**

Strengths
- The paper proposed a novel modification of a GP able to handle lower bound information and proved its higher expressiveness compared to a classic GP.
- The paper proposed an adaptation to the EI acquisition criterion that incorporates lower bound information; these adapted acquisition functions can be used with both SlogGP and classic GP.
- The proposed method is shown to outperform classic GP in the considered experiments, which may motivate wider adoption of SlogGP (pending further investigation in the appropriate use cases). This would represent a significant impact to the field.
- The paper provides a set of ablation studies to evaluate various components of the model.
- The paper is well-written and clear.

Weaknesses
- There is limited discussion on what it means when estimated ζ is far away (or lower) than the known lower bound (it is only touched upon at the end of *Conclusion*).
- Section 3.4. has many highly specific steps that would benefit from providing more intuition.
- The latter benchmarking sections (starting from 5.1) become less clear and appear more rushed in presentation.

---

> ### Author Response · Authors · 2025-01-17
> **Response to Reviewer RefA Comments**
>
> We appreciate the constructive feedback. Due to space limitations, we respond only to the most important questions and the original comment is shortened. We will modify the paper accordingly, although this will take a bit more time.
>
> __State if there are any commonly used conditions under which Theorem 3.1 does not hold. Can a SlogGP work in a noisy setting?__
>
> The mean function of the target GP has to be a constant. In a setting with normally distributed noise, observations would be unbounded, and thus it is not clear whether BABO would be beneficial. We therefore restrict our considerations to the noiseless case.
>
> __More intuition for why U is multiplied by the provided expression and define ϕ−1 after estimating MLE of ζ.__
>
> The ζ_prior follows a shifted log-normal distribution, characterized by an underlying normal distribution N(μ,σ^2) and a shift parameter C. For any value ζ^, its probability can be computed by recognizing that ln(ζ^-C) follows N(μ,σ^2).
> We detect a prior conflict when ζ^​ is highly unlikely under the prior distribution, i.e., when ln(ζ^-C) is highly unlikely under N(μ,σ^2). Then we scale ϕ^{−1} to increase σ^2, ensuring that ζ^​ is more likely under the adjusted prior distribution.
>
> __Why does signal variance being too small indicate the data-prior conflict? Is it because that would happen when  ζ→inf, which would indicate it being far away from fb?__
>
> Yes, according to Theorem 3.1, ζ→∞ as the signal variance approaches 0.
>
> __How were δ1, δ2, δ3 chosen?__
>
> Please see our response to Reviewer oC82.
>
> __Please show why SlogTEI becomes SlogEI for −ζ>fb.__
>
> Because we truncate any value below fb, but when −ζ>fb, i.e. the lower bound of the SlogGP model is higher than fb, all values are above fb anyway, and SlogTEI becomes SlogEI.
>
> __Fig. 7: Which method estimated the ζ^shown with the "no bound information" curve?__
>
> This refers to SlogGP+SlogEI. Since there is no bound information, we use MLE to estimate ζ.  We will incorporate this information and include the known lower bound value in the plots.
>
> __Fig 7: "No bound information" curve often converges to a lower bound than BABO. Did it find a better optimum? In most scenarios, both methods converge to the same value; what is the interpretation?__
>
> Yes, the method with bound information is BABO. These plots only show the evolution of the parameter -ζ, they don't say anything about the quality of the optimum. But a lower -ζ value means being closer to ignoring any bound. The prior bound information is only a prior, so eventually we would expect the data to determine ζ, with informed prior or not.
>
> __Provide a discussion on the difference between ζ being far away from fb and model mismatch and how this connects to (i) BABO with fixed ζ performing worse than having to learn ζ, (ii) BABO learning estimated ζ that is lower than fb (e.g. Ackley in Fig.7) and (iii) absorbing both skewness and lower bound information into a single parameter.__
>
> If ζ is far away from fb, we have a model mismatch between the SlogGP model using the prior bound information which is skewed close to the bound, and the data. A standard GP would be the better model. SlogGP then ignores the prior bound information and learns a large ζ, which makes SlogGP similar to a standard GP. With a fixed ζ, the algorithm doesn’t have this option, hence is performing worse.
> Skewness and lower bound are closely linked - without skewness, no bound information in the model (but still in the acquisition function, hence the importance of SlogTEI).
>
> __The previous statement suggests that this wouldn't hold on Ackley, yet BABO still performs best.__
>
> This may be due to numerical issues. When optimizing the Ackley function, SlogGP learns a very small signal variance (approximately 1e-5), and the resulting noise variance of 1e-10. However, we observe that the effective noise (due to numerical issues) ends up to be around 1e-6. This seems beneficial for the Ackley function, as it helps to handle Ackley's numerous local optima. This phenomenon is specific to Ackley due to the very small learned signal variance.
>
> __Tab 1: Why is SlogGP error on the SlogGP test function higher than SlogGP on the GP test function?__
>
> This is because a SlogGP-generated function has a larger scale.
>
> __Fig. 8a: Both GP and SlogGP using TEI perform worse than all other methods.__
>
> Since the max_depth parameter in XGBoost is integer-valued, this creates discontinuities in the objective landscape. The results could be attributed to model mismatch.
>
> __Can other acquisition functions be modified to be used in SlogGP?__
>
> Commonly-used acquisition functions such as PI or MES can be modified to be used with SlogGP.
>
>  __How often is the lower bound information available in practice?__
>
>  Bound information is quite common since many problems naturally have trivial bounds (e.g., accuracy<1, distance>0). The usefulness of these bounds depends on the scale of the bounds relative to the problem.

---

> > ### Comment · Reviewer_RefA · 2025-02-02
> > **Response to authors**
> >
> > Thank you for the clarifications and the updated revision.
> >
> > Remaining questions:
> > 1. I am still uncertain how to interpret the estimated bound plots in Fig. 8 and 10. I understand that those figures show $\zeta$ estimated with SlogGP+SlogEI (no prior on the bound) and BABO (prior on the bound included). However, the caption of Fig. 8 compares the performance of SlogGP and GP. Are these plots informative insofar as they show if $\zeta$ is far or close to the true lower bound? If yes, then what additional information does "no bound" curve provide? Below I put more specific questions for clarification.
> >
> > > For functions like Ackley and Hartman, where bounds are highly negative relative to the objective range, SlogGP behaves similarly to GP since the lower bound provides little guidance.
> >
> > Rosenbrock and Powell also have highly negative bounds relative to the objective range, but BABO (and other SlogGPs) outperform GPs on those as shown in Fig. 7.
> >
> > > However, for functions like Beale where the estimated bound converges near the true minimum, this additional bound information proves more beneficial.
> >
> > Does the benefit of bound information refer to the gap between bound/no bound curves during iterations in Fig. 8 or SlogGP outperforming GP in Fig. 7?
> >
> > >The proximity between this estimate and our known lower bound (fb := 0%) enables rapid convergence to the optimal lower bound, enhancing BO performance.
> >
> > Does this refer to the bound curve (BABO, green) converging quicker than the no-bound curve (SlogGP+SlogEI curve, red)? if yes, why is it cited as supporting evidence for SlogGP outperforming GP?
> >
> > >This proximity accelerates ˆζ learning and improves overall BO performance. ...enabling rapid SlogGP model learning and consequently enhancing BO performance.
> >
> > Similar questions as above about statements made about Robot Push and Beale in Section 5.1.
> >
> > 2. In Section 3.4, are $\mathbb{P}$ and $\phi$ supposed to be CDFs? If not, can you please elaborate on what $|\phi^{-1}(\mathbb{P}(.))|$ is computing. For example, a standard normal PDF has two corresponding $x$ locations for every value except the mode so I'm unclear how inverse mapping would work.
> >
> > Suggestion:
> > - I think incorporating the explanations for the $\delta$ hyperparameters (provided at the beginning of Section 5.2) into their first occurrences in the main text (Section 3.4) would help the readers gain a better intuition for them early on.
> >
> > Minor:
> > - p.13: "... mechanism automatically adjust the prior" -> "... adjusts"
> > - Fig. 18 and 19 should have more informative captions
> > - p. 8: "... setting a universal threshold" -> "however, setting"
> > - Cite the first occurrence of the PDE Variance problem (p. 10)
> > - p.13: The noise variance used for numerical stability is not discussed anywhere in the main text (only in the Appendix). It should be briefly introduced in the main text and the reader should be referred to the relevant Appendix section

---

> > > ### Author Response · Authors · 2025-02-07
> > > **Response to Reviewer RefA Comments**
> > >
> > > Thank you for your follow-up questions.
> > >
> > > __Are these plots informative insofar as they show if  ζ is far or close to the true lower bound? If yes, then what additional information does "no bound" curve provide?__
> > >
> > > Thank you for your constructive feedback. After reconsidering, we concluded that perhaps the estimated lower bound (Fig. 8 and Fig. 10) does not really provide additional insights.
> > >
> > > Our initial intention was to illustrate how the estimated lower bound helps explain two key points:
> > >
> > > (1) SlogGP + SlogEI outperforms GP + EI: When the estimated lower bound is not highly negative, SlogGP should have an advantage over GP, as it can better capture skewness and account for the lower bound. Consequently, SlogGP + SlogEI is expected to perform better than GP + EI.
> > >
> > > (2) SlogGP$^b$ + SlogEI compared to SlogGP + SlogEI: When the estimated lower bound converges to the true lower bound, prior knowledge of the true lower bound should be more beneficial. The relationship here is less well-supported, making the conclusion less robust.
> > >
> > > There are two reasons why Fig. 8 and Fig. 10 have limited explanatory power. One is that a different estimated bound will influence data collection, which in turn influences the estimated bounds, making it difficult to discern the influence of a particular factor. The other is that the variability of the lower bound estimates is quite large, in most cases the differences are not statistically significant.
> > > As a result, we have decided to remove the two figures and the discussion.
> > >
> > > That said, we still want to address a few of your questions.
> > >
> > > __For functions like Ackley and Hartman, where bounds are highly negative relative to the objective range, SlogGP behaves similarly to GP since the lower bound provides little guidance. Rosenbrock and Powell also have highly negative bounds relative to the objective range, but BABO (and other SlogGPs) outperform GPs on those as shown in Fig. 7.__
> > >
> > > Yes, the estimated lower bound is very negative for Ackley and Hartman, but for Rosenbrock and Powell, considering their range is large, the estimated bounds are not that negative for them. For instance, the Powell function's range spans from $0$ to the order of $10^4$, making an estimated bound of $-10^3$ relatively reasonable. The key insight is that the significance of a lower bound must be evaluated relative to the function's range rather than its absolute value alone.
> > >
> > > __In Section 3.4, are P and ϕ supposed to be CDFs? If not, can you please elaborate on what |ϕ−1(P(.))| is computing. For example, a standard normal PDF has two corresponding x locations for every value except the mode so I'm unclear how inverse mapping would work.__
> > >
> > > Thank you for your observation. We acknowledge there was an error in the formula. We use the absolute value of the standard score (the z-value in statistics) to determine whether there is a mismatch between the prior distribution and the estimated zeta.
> > >
> > > The prior follows a shifted log-normal distribution: $\zeta \sim \exp(N(\mu,\sigma^2))-C$. More specifically, $\ln(\zeta+C) \sim N(\mu,\sigma^2)$. Consequently,  the absolute value of the standard score for the estimated $\hat\zeta$ is $\left|\frac{\ln(\hat{\zeta}+C)-\mu}{\sigma}\right|$. A large absolute value indicates a prior-data mismatch, prompting us to reduce reliance on the prior by increasing the variance in subsequent steps.
> > >
> > > We have revised this part in our manuscript.
> > >
> > > __Suggestion and Minor__
> > >
> > > We have refined the manuscript based on the reviewers' suggestions. A revised version has been uploaded, with the changes highlighted in red.

---

> > > > ### Comment · Reviewer_RefA · 2025-02-13
> > > > **Response to authors**
> > > >
> > > > Thank you for your response. I spotted a typo: an extra "." at the end of the added "...that sets the threshold for detecting prior conflicts.)" on p. 8. and inconsistent spelling of "log-normal".

---

> > > > > ### Author Response · Authors · 2025-02-14
> > > > > **Response to Reviewer RefA Comments**
> > > > >
> > > > > Thank you for your careful observation. The typos have been fixed.

---

### Author Response · Authors · 2025-01-23
**Revision**

Dear Reviewers,

Thank you for your valuable feedback. Based on your suggestions, we have updated our paper with the following key improvements:

1. Added more real-life examples in Introduction

2. Added the mean and the variance of SlogGP in Section 3.1

3. Relocated SlogEI calculation to Section 3.3

4. Refined explanations of $\delta_1$, $\delta_2$, and $\delta_3$ in Section 3.4

5. Expanded XGBoost test problems and added PDE Variance problems in Sections 4.2 and 4.3

6. Added hyperparameter sensitivity analysis in Section 5.2

7. Added visualization of SlogGP in Appendix A.3

8. Corrected typos

9. Made the x-label and y-label larger in figures and added the known bound $f^b$ in lower bound plots

Also, we highlighted the modified sections in blue for easier identification.

Please note that this is work-in-progress, and our intent is to provide you with an early view of the revision. We hope this gives you insight into the progress we are making and invites further valuable feedback. Thank you.

---

### Decision · Action_Editor_wavw · 2025-02-27

**Recommendation:** Accept as is

**Comment:**

The paper is about Bayesian optimization when knowledge about the optimal value is available. The authors provide a method to incorporate such knowledge into a new surrogate model (based on shifted log-normal GPs) and derive the expected-improvement acquisition function for their model.

The reviewers all agree that the paper should be published. I agree. The paper is well crafted with a substantial amount of empirical evidence.

**Audience:**

The paper is on Bayesian optimization which is of interest to TMLR's audience.

**Claims And Evidence:**

The claims made in the paper are supported by analytical results and empirical comparison with other relevant methods on a range of benchmarks.